# Is Oracle Pruning the True Oracle?
# – A Sanity-Check of Neural Network Pruning with Retraining

**Sicheng Feng**\*                                                                          *fengsicheng@u.nus.edu*
*Westlake University, Hangzhou, China*
*Nankai University, Tianjin, China*

**Keda Tao**                                                                                *taokeda@westlake.edu.cn*
*Westlake University, Hangzhou, China*

**Huan Wang**†                                                                            *wanghuan@westlake.edu.cn*
*Westlake University, Hangzhou, China*

**Reviewed on OpenReview:** *https://openreview.net/forum?id=k13YnckIZY*

https://fscdc.github.io/Oracle-Pruning-Sanity-Check

## Abstract

*Oracle pruning*, which selects unimportant weights by minimizing the pruned train loss, has served as the foundation for most neural network pruning methods for over thirty-five years, while few (if any) have thought about how much the foundation really holds. This paper, for the first time, attempts to systematically examine its validity on deep neural networks through empirical correlation analyses and provides meta-framework reflections on the field of neural network pruning. Specifically, this paper focuses on the pruning algorithms with three stages: training, pruning, and retraining. We analyze the correlation in model performance before and after the retraining stage. Extensive experiments (**37K** models are trained) across a wide spectrum of models (LeNet5, VGG, ResNets, ViT, MLLM) and datasets (MNIST, CIFAR10/CIFAR100, ImageNet-1K, MLLM data) are conducted. For large-scale experiments, we adopt approximate oracle pruning due to the prohibitive cost of exact oracle pruning. The results point to a counterintuitive conclusion: for deep learning models of nontrivial size (already at the scale of ResNet56 on CIFAR-10), pre-retraining performance is *negligibly correlated* with post-retraining performance. In other words, the weights identified by oracle pruning can *scarcely* guarantee strong performance following retraining. This further suggests that existing works that derive pruning criteria from oracle pruning may rest on a questionable foundational premise. Further studies suggest that rising task complexity is a primary factor behind the invalidity of oracle pruning nowadays. Finally, given the evidence, we argue that the retraining stage in a pruning algorithm should be accounted for when developing pruning criteria.

## 1  Introduction

Neural network pruning removes less important parameters from a large network to find the appropriate model size (Baum & Haussler, 1988; Hanson & Pratt, 1988) or improve the model efficiency (*e.g.*, smaller model, faster inference) (Cheng et al., 2017; Deng et al., 2020). The topic has been studied for over thirty-five years, even before the current deep learning era (Schmidhuber, 2015; LeCun et al., 2015) of AI.

A pruning algorithm typically consists of three steps (see illustration in Fig. 1): (1) training, which trains the original dense model; (2) pruning, which removes parameters from the dense model based on specific

---

\*Work done when Sicheng was a visiting research intern at ENCODE Lab, Westlake University.
†Corresponding author.

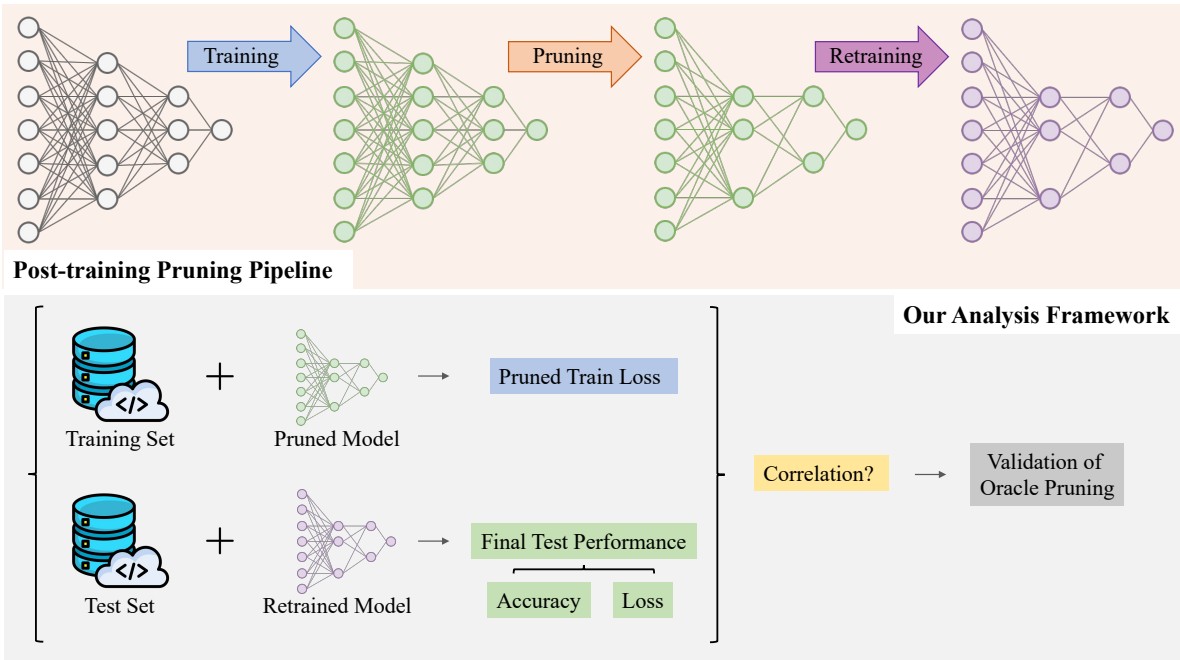

Figure 1: **Analysis framework of this work.** We study the validity of oracle pruning in this paper, by examining the correlation between the pruned train loss and the final test performance (*i.e.*, test accuracy or test loss). We apply this analysis framework to a wide range of networks and datasets (from toy networks like LeNet5-Mini to large ones like TinyLLaVA-3.1B) in order to have a comprehensive evaluation. The key finding of this work is that, to our surprise, oracle pruning becomes invalid on modern networks and datasets (starting from the CIFAR level), challenging the conventional belief in network pruning for the past 35 years.

criteria; and (3) retraining, which retrains the pruned model to recover performance. This three-step process (post-training pruning) has been practiced for over thirty-five years (Mozer & Smolensky, 1988; Baum & Haussler, 1988; Chauvin, 1988; Karnin, 1990) and is still widely adopted in modern pruning methods (Hoefler et al., 2021; Sze et al., 2017; Fang et al., 2024; Zhu et al., 2026).

Since the inception of pruning, most research has focused on the second step, determining which weights to remove, which is known as the weight importance scoring (or pruning criteria) problem. For weight importance scoring, oracle pruning (Mozer & Smolensky, 1988; LeCun et al., 1990) is a very straightforward and fundamental[1] methodology for identifying and removing the least important parameters: it removes the weights whose removal incurs the least increased error, which can be formulated as follows,

$$\min_{\mathcal{M}} \left( \mathcal{L}(\mathcal{D}|\mathcal{W}') - \mathcal{L}(\mathcal{D}|\mathcal{W}) \right), \mathcal{W}' = \mathcal{W} \odot \mathcal{M}, \quad s.t. \ |\mathcal{M}|_0 = C, \tag{1}$$

where $\mathcal{D}$ is the training set, defined as $\mathcal{D} = (\mathcal{X}, \mathcal{Y})$, $\mathcal{X} = \{x_0, x_1, \ldots, x_N\}$ represents the inputs; $\mathcal{Y} = \{y_0, y_1, \ldots, y_N\}$ represents the targets; $\mathcal{W}$ represents the original network parameters; $\mathcal{W}'$ represents the pruned network parameters. The pruning is implemented as an element-wise product between the original network $\mathcal{W}$ and the mask $\mathcal{M}$. $C$ is usually predefined as the number of nonzero parameters.

A naive way to implement oracle pruning is by exhaustive search: Try every possible mask and record the loss change; choose the mask that increases the least loss. Obviously, this is not practical due to the large number of pruning combinations, so many follow-up works use various approximation methods to approximate Eq. (1) at the model or layer level (He et al., 2017; Jiang et al., 2018). One prevailing idea in the literature is to use the Taylor series to expand Eq. (1) (with approximations) and truncate the series after the second-order

---

[1]The idea can date back to at least the 1980s (Mozer & Smolensky, 1988; LeCun et al., 1990), and is still widely adopted as the basis in many very recent pruning papers such as Ma et al. (2023); Kim et al. (2024); Fang et al. (2024).

term,

$$\delta\mathcal{L} = \mathbf{G}^\top \delta\mathbf{W} + \frac{1}{2}\delta\mathbf{W}^\top \mathbf{H}\,\delta\mathbf{W} + \mathcal{O}(\|\delta\mathbf{W}\|^3), \tag{2}$$

where $\delta\mathcal{W}$ denotes the parameter perturbation induced by pruning, $\mathbf{G} = \nabla_\mathcal{W}\mathcal{L}$ is the gradient of the loss with respect to the model parameters, and $\mathcal{H} = \nabla_\mathcal{W}^2\mathcal{L}$ is the corresponding Hessian matrix. The higher-order term $\mathcal{O}(\|\delta\mathcal{W}\|^3)$ is typically omitted due to its computational intractability.

Many works (Mozer & Smolensky, 1988; LeCun et al., 1990; Karnin, 1990; Molchanov et al., 2017; 2019; Lee et al., 2019; Wang et al., 2019a) are based on the above idea, or its variants (Tartaglione et al., 2018; Yu et al., 2018; Ding et al., 2019; Fang et al., 2024). The oracle pruning idea has also found extension usage in compressing non-classification models, such as text-to-image diffusion models (Kim et al., 2024; Zhu et al., 2026) and large language models (Ma et al., 2023; Fang et al., 2023).

Despite the long history and extensive usage, few (if any) have questioned whether the idea of oracle pruning really holds. In this paper, we think we should reexamine its validity, at least for modern deep learning models, for the following reasons: (1) Several recent studies (Gale et al., 2019; Li et al., 2022; Wang et al., 2023) have reported a puzzling phenomenon that the simple magnitude pruning (Han et al., 2016; Li et al., 2017) or even random pruning (Li et al., 2022) can match or even surpass many pruning criteria derived from the Taylor expansion form of oracle pruning. Namely, the theories appear sound, while the promised methodology advantage of oracle pruning is never fulfilled in practice; (2) A limitation in different pruning criteria noted by (Huang et al., 2021) is that different pruning approaches actually select very similar weights to remove. Many widely cited pruning criteria yield nearly identical importance scores for filters, leading to similar pruned network structures, despite theoretical differences among the criteria. This overlap suggests that certain pruning criteria may lack distinctiveness in practice.

These counterintuitive empirical observations have confounded researchers for quite a while (Gale et al., 2019; Blalock et al., 2020; Wang et al., 2023), but no systematic investigation has been done to explain them. In this paper, after we present the new evidence that oracle pruning, which is the foundation of many pruning criteria in the conventional three-stage pruning pipeline, turns out to be *not so grounded*, those "mysterious" observations will become easy to comprehend.

Specifically, we examine the validity of oracle pruning by analyzing the statistical correlation between the pruned model performance (measured by the *pruned train loss*) and the final model performance (measured by the *test accuracy or loss*) after **retraining**. The analyses are conducted on a wide range of models and datasets with **37K** models trained, from the toy-level models like LeNet5-Mini (LeCun et al., 1998) to more recent VGG19 (Simonyan & Zisserman, 2015), ResNets (He et al., 2016), and attention-based Vision Transformers (ViTs) (Vaswani et al., 2017; Dosovitskiy, 2020), and a multimodal large language model (MLLM) TinyLLaVA-3.1B (Zhou et al., 2024). To our surprise, the results suggest the pruned train loss actually poses a very weak (if any) correlation with the final test performance after retraining. Namely, the idea of oracle pruning does not hold. Further results suggest that the rising task complexity is a key factor making oracle pruning invalid, compared to the 1980s when oracle pruning was almost true (due to the simple nature of problems at that time).

Our contributions in this work are as follows: (1) Oracle pruning is extensively taken as the basis for many pruning algorithms. Its validity is of great significance, but has not been systematically studied for deep neural networks. This paper fills the gap; (2) Methodologically, we present an analysis framework based on Kendall correlation (Sec. 3.1), and two proposed metrics (anomaly ratio, counterexample ratio; Sec. 3.2) to examine the validity of oracle pruning, which can also be used to evaluate the validity of other pruning criteria under the retraining paradigm; (3) Empirically, we conduct extensive experiments (37K models are trained) to analyze the correlation between the pruned train loss and final test performance. The results, to our surprise, suggest that the pruned train loss poses weak-or-none correlation *w.r.t.* the final test performance, challenging the validity of oracle pruning (Sec. 3.3); (4) We present further evidence to show that it is the increasing task complexity (such as more challenging datasets, and more complicated networks) that renders oracle pruning ineffective (Sec. 4).

## 2 Related Work

Researchers commonly classify pruning methods based on three key factors: (1) Base model – this refers to the timing of pruning, *i.e.*, whether pruning is applied to a pre-trained model or a randomly initialized model; (2) Sparsity granularity – this defines the smallest unit or group of weights that can be pruned; and (3) Pruning criterion – this determines the metric or method used to differentiate important weights (those to be retained) from unimportant ones (those to be pruned). In the following section, we elaborate on these three dimensions and provide the necessary background for understanding various pruning approaches. Additionally, we add a brief discussion about pruning during training.

**Pruning after training *vs.* pruning at initialization.** Traditionally, pruning has been predominantly applied to pre-trained models, a process referred to as post-training pruning (i.e., training-pruning-retraining). This approach, which involves training a full model before selectively removing less important weights, has long been the standard practice in the field. However, more recent research has introduced the idea of pruning at initialization, where pruning is conducted on a randomly initialized model rather than a fully trained one. Methods such as SNIP (Lee et al., 2019) and the Lottery Ticket Hypothesis (Frankle & Carbin, 2019) have demonstrated that pruning during the early stages of training can yield competitive performance, potentially matching that of dense models. While pruning at initialization (Frankle et al., 2021; Lee et al., 2020; Ramanujan et al., 2020; Wang et al., 2020) presents a promising alternative, it is less relevant to this paper, which focuses on the complexities and challenges associated with post-training pruning. Comprehensive discussions on initialization-based pruning can be found in related reviews (Wang et al., 2022), but this paper centers on evaluating and improving the traditional pruning techniques applied to pre-trained networks.

**Structured pruning *vs.* unstructured pruning.** Network pruning can be classified into structured pruning (Li et al., 2017; Wen et al., 2016; He et al., 2017; 2018; Wang et al., 2022) and unstructured pruning (Han et al., 2015; 2016; LeCun et al., 1990; Hassibi & Stork, 1993; Singh & Alistarh, 2020), depending on the sparsity structure. Structured pruning focuses on removing entire structures, such as filters or channels, to reduce computational overhead and improve inference speed. In contrast, unstructured pruning removes individual weights, leading to a sparse network, which reduces the model size but offers less practical speedup. This paper mainly focuses on structured pruning, specifically filter pruning, as the primary goal with modern networks, such as ResNets (He et al., 2016), is to enhance inference speed rather than simply reducing the model size, which was a more significant concern. Besides, we also discuss unstructured pruning on MLLM when exploring the phenomena mentioned above.

**Importance-based pruning *vs.* regularization-based pruning.** In this axis, two primary ways are widely used to determine which weights to remove: importance-based and regularization-based pruning. Importance-based methods prune weights based on specific criteria, such as weight magnitude for unstructured pruning (Han et al., 2016; 2015) or $L_1$-norm for filter pruning (Li et al., 2017). These methods can also incorporate second-order gradient information, such as the Hessian or Fisher matrix (Hassibi & Stork, 1993; LeCun et al., 1990; Singh & Alistarh, 2020; Theis et al., 2018; Wang et al., 2019a), to assess the saliency of weights. Regularization-based approaches introduce a penalty term to the objective function, encouraging unimportant weights to move toward zero and then prune weights with the smallest magnitudes. Notably, regularization-based methods often still rely on importance measures during the final pruning stage. Although these two paradigms are sometimes combined (Ding et al., 2018; Wang et al., 2021; 2019b), our work focuses on importance-based pruning. Specifically, we investigate one-shot pruning, where unimportant weights are pruned in a single step rather than through iterative processes. This approach is advantageous for reducing model complexity with minimal computational overhead, as it allows for efficient pruning without the need for extensive fine-tuning after multiple pruning rounds.

**Pruning during training.** There is also a line of work that incorporates sparsification directly into the training process. Srinivas & Babu (2016) introduced generalized dropout, which learns per-unit dropout probabilities during training and can yield sparse networks after convergence, while Louizos et al. (2017) proposed an alternative training-time sparsification approach based on optimizing L0-regularized networks via continuous relaxations to address the discontinuity of the L0 norm. More recently, Rajpal et al. (2023) introduced a Bayesian early-pruning framework that models the efficacy of network components during training and prunes low-utility elements accordingly. These approaches differ from the post-training pruning

setting considered in this work. Our focus is on pruning pre-trained models, which remains the primary target of network pruning methods.

## 3 Examining Oracle Pruning

In this section, we systematically study the effectiveness of oracle pruning by analyzing the correlation between pruned train loss and final test performance (see Fig. 1). In the following subsections, we first introduce the analysis methods, then we present and analyze the results across multiple networks and datasets.

### 3.1 Framework of Correlation Analysis

Given a model and a dataset, we first train the model to convergence to obtain the pre-trained model. Then we conduct structured pruning (*i.e.*, some filters of the model are removed), and then retrain the pruned model to regain performance. To obtain fair and representative results, some key pruning details need to be properly determined, as shown below.

**How many to prune?** We prune each model with a uniform layerwise pruning ratio. The pruning ratio should not be extreme (too small or too large). Without any prior, we choose 50% layerwise sparsity (unless we aim to see the effect of varying pruning ratios, such as Fig. 2 and Tab. 2).

**Which to prune and how to realize oracle pruning?** We intentionally include some small networks, on which we can realize oracle pruning exactly (*i.e.*, without any approximation), by exhaustively searching all the pruning combinations. For practical models, the exhaustive search is not possible; so we randomly sample abundant (*e.g.*, 1K) pruning combinations to calculate the correlation. Additionally, we state that our target is not to identify the true oracle pruning point (*i.e.*, the one with least training loss) but to sample enough pruning combinations for correlation analysis.

**Correlation between what?** We analyze the correlation between **pruned train loss** and **final test performance** (shown in Fig. 1). The pruned train loss refers to the loss evaluated on the training set after the model has been pruned, without retraining, and also serves as the pruning criterion under the oracle pruning assumption. In contrast, the final test performance denotes the performance of the model on the testing set after retraining, measured specifically by test accuracy and test loss.

**What correlation metric is used?** We employ Kendall correlation (Kendall, 1948; Freedman & Pisani, 1998; Johnson et al., 2002). In statistics, three correlation analysis methods are widely used: Pearson, Spearman, and Kendall. Pearson is usually used for measuring linear correlation, the other two for non-linear correlation. Between Pearson and Kendall, Kendall directly counts how many pairs agree or disagree in their rankings, which is more applicable to our case, so we choose Kendall. The formal definition of the Kendall coefficient $\tau$ is a non-parametric measure of correlation, which evaluates the ordinal association between two variables. The Kendall coefficient $\tau$ is defined as:

$$\tau = \frac{C - D}{\frac{1}{2}n(n-1)}, \tag{3}$$

where $C$ is the number of concordant pairs, where both variables change in the same direction (either both increase or both decrease); $D$ is the number of *discordant* pairs, where one variable increases while the other decreases; $n$ is the total number of observations. The Kendall coefficient $\tau \in [-1, 1]$, where $\tau = 1$ indicates perfect agreement, $\tau = -1$ perfect disagreement, and $\tau = 0$ independence between the two rankings.

In addition to the Kendall coefficient $\tau$, we also report the p-value to show how significant the correlation is. By convention, a p-value less than 5% is considered statistically significant. In our case, if we expect the pruned train loss can help us select the model with a good final test loss, the $\tau$ should be noticeably positive with a p-value less than 5%. In short, the validity of oracle pruning is defined as follows.

Table 1: Summary of the analysis methods for checking the validity of oracle pruning on different networks and datasets. LeNet5-Mini is a small enough network to achieve oracle pruning exactly; for other networks, we randomly sample enough models to analyze Kendall correlation, anomaly ratio, and counterexample ratio.

| Network (Dataset) | Kendall correlation | Anomaly ratio | Counterexample ratio |
|---|---|---|---|
| LeNet5-Mini (MNIST) | ✓ | ✓ | ✗ |
| ResNet56 (CIFAR10) | ✓ | ✓ | ✗ |
| VGG19 (CIFAR100) | ✓ | ✓ | ✗ |
| ResNet18 (ImageNet-1K) | ✓ | ✓ | ✗ |
| ViT-B/16 (ImageNet-1K) | ✓ | ✗ | ✓ |
| TinyLLaVA-3.1B (Five benchmarks) | ✗ | ✗ | ✓ |

**Definition 3.1** (Validity of Oracle Pruning). Oracle pruning is considered valid only when $0.2 < \tau \leq 1$ (*i.e.*, for the correlation between pruned train loss and final test loss) or $-1 \leq \tau < -0.2$ (*i.e.*, for the correlation between pruned train loss and final test accuracy), and p-value is less than 5%. For all the other cases, oracle pruning is considered invalid.

## 3.2 Supplementary Analysis Metrics: Anomaly Ratio & Counterexample Ratio

In addition to the correlation coefficient, we also consider other metrics that can help us assess the effectiveness of a pruning criterion: anomaly ratio & counterexample ratio. The details are as follows.

**Anomaly ratio (oracle pruning *vs.* random pruning).** Random pruning is the baseline of all pruning criteria. Comparison with it works as a sanity check for any pruning criterion. After we obtain the scatter points of pruned train loss and final test accuracy (or loss), we can count how many samples (denoted as $N_a$) have a better final test accuracy (or loss) than the sample selected by oracle pruning[2]. The ratio of $N_a$ over the total number of samples (denoted as $N_{total}$) is defined as anomaly ratio,

$$r_{\text{anomaly}} = \frac{N_a}{N_{\text{total}}}. \tag{4}$$

If a pruning criterion is considered valid, the anomaly ratio should be noticeably smaller than 50%. Otherwise, it means that simply by random sampling, it is considerably probable to obtain a better result than using the proposed pruning criterion, *i.e.*, the pruning criterion is meaningless.

**Counterexample ratio.** A counterexample is defined as two pruning combinations where one combination has a lower pruned train loss but results in a worse final test accuracy (or loss). This is considered a counterexample because when a pruning combination has a lower pruned train loss, we expect it to perform better after retraining. If the opposite occurs, it constitutes a counterexample. The counterexample ratio is the ratio of the number of counterexamples over the total number of pairs in pruning combinations. We introduce this metric because on some large networks (*e.g.*, the recent large language models), we only have a handful of experiments, not enough for rigorous correlation analysis.

In this part, if we consider a pruning criterion based on minimizing the pruned train loss to be effective, the counterexample ratio should be significantly smaller than 50%.

## 3.3 Results for Examining Oracle Pruning

### 3.3.1 Experiment Settings

**Networks and datasets.** We evaluate oracle pruning on a wide range of networks and datasets. We first conduct experiments on the MNIST dataset (LeCun et al., 1998) with LeNet5-Mini, a simplified version of LeNet5 (LeCun et al., 1998) with 10 filters or neurons in the convolutional (conv) or fully connected layers. Then we follow existing works (Ding et al., 2021; Wang et al., 2021) to conduct experiments on larger-scale datasets with standard convolutional networks: ResNet56 (He et al., 2016) on CIFAR10 (Krizhevsky,

---

[2]For cases where we cannot traverse all pruning combinations, we selected enough samples to approximate oracle pruning.

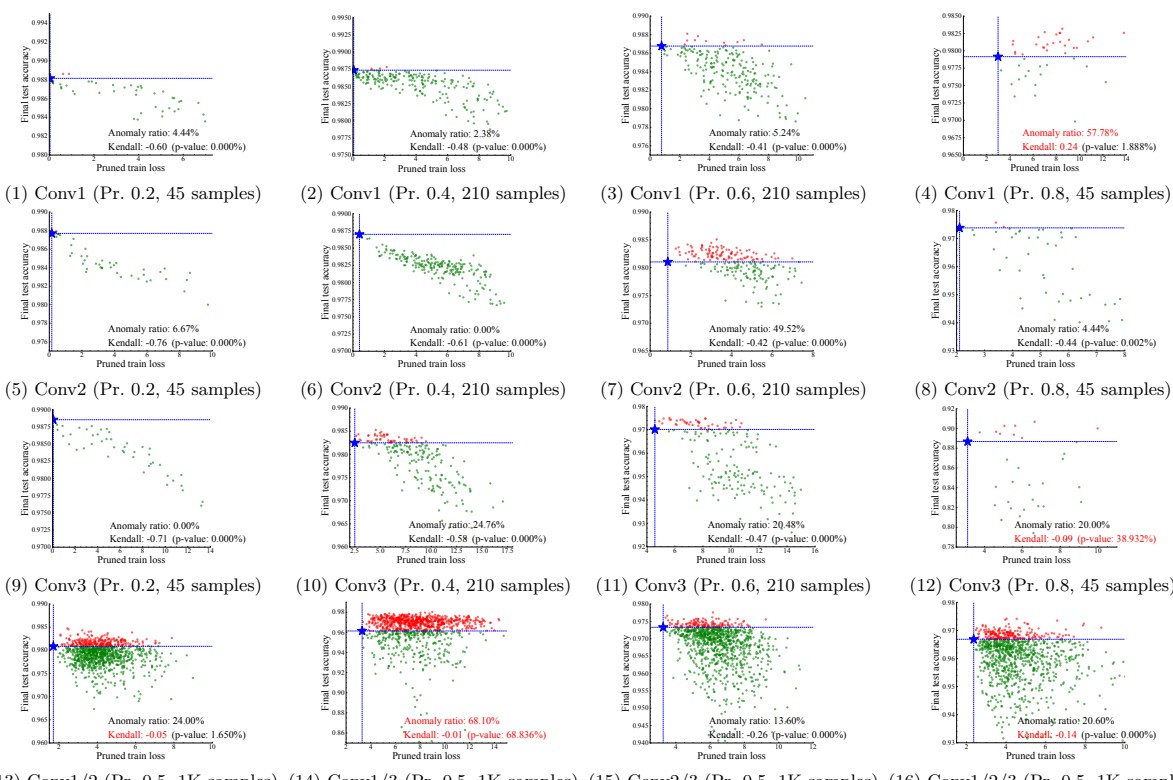

Figure 2: Pruned train loss *vs.* final test accuracy on MNIST with LeNet5-Mini. The subcaptions correspond to the pruning settings. The blue star indicates the *reference oracle pruning result* (*i.e.*, the one with the smallest pruned train loss). The points with final test accuracy higher than the reference oracle pruning result are marked in red (anomaly points), and those lower are marked in green.

2009), VGG19 (Simonyan & Zisserman, 2015) on CIFAR100 (Krizhevsky, 2009), ResNet18 (He et al., 2016) on ImageNet-1K dataset (Deng et al., 2009). We further experiment on ViT (Dosovitskiy, 2020) with the ImageNet-1K dataset (Deng et al., 2009). We further have experiments with MLLM, specifically, TinyLLaVA-3.1B (Zhou et al., 2024). More Details are shown in Appendix A.1.

**Settings.** For the pruning ratio, we use our specified ones. After pruning, all pruned models will be retrained with typical and proper configurations[3]. We further provide retraining settings in Appendix A.2 and pruning ratio settings in Appendix A.3. A summary of the analysis metrics is shown in Tab. 1.

**Remarks.** The LeNet5-Mini network appears "toy" and may be considered negligible for modern pruning. Yet here, we encourage the readers to refrain from this thought first. We intentionally designed the experiments on this network because we can achieve oracle pruning exactly. Analysis results on this network (*e.g.*, Tab. 2) can help us understand when oracle pruning starts to turn ineffective. Additionally, we extend our analysis framework to other pruning methods, such as magnitude pruning and Taylor-FO pruning (Molchanov et al., 2017), and report the corresponding results in Appendix B.1.

### 3.3.2 Results with LeNet5-Mini on MNIST

**Kendall correlation.** As seen in Tab. 2: (1) when pruning one layer, as the pruning ratio increases from 0.2 to 0.8, the correlation between pruned train loss and final test accuracy weakens, indicated by the smaller absolute Kendall coefficient. This trend is generally consistent across different layers (Conv1 / Conv2 / Conv3). In some cases, the correlation can turn wrong, *e.g.*, for Conv1 layer, pruning ratio 0.8, the correlation

---

[3]Previous works (Renda et al., 2020; Wang et al., 2023) noticed that the learning rate in the retraining stage is critical to the final performance. We are aware of this and have used the right hyperparameters to ensure the model is fine-tuned properly.

Table 2: Kendall correlation between pruned train loss and final test accuracy, by exhaustively pruning LeNet5-Mini network on MNIST dataset. Each entry in the table is arranged as Kendall coefficient / p-value. *Pr.* means the pruning ratio for the corresponding layer combination of Conv1, Conv2, and Conv3. The red indicates the results where oracle pruning is invalid as defined in Sec. 3.1.

| Pr. | Conv1 | Conv2 | Conv3 |
|---|---|---|---|
| 0.2 | -0.60 / 4.9e-09 | -0.76 / 2.0e-13 | -0.71 / 7.0e-12 |
| 0.3 | -0.51 / 1.3e-16 | -0.61 / 4.8e-23 | -0.59 / 9.8e-22 |
| 0.4 | -0.48 / 1.3e-24 | -0.61 / 7.4e-39 | -0.58 / 2.2e-35 |
| 0.5 | -0.51 / 2.1e-33 | -0.51 / 1.8e-33 | -0.60 / 2.5e-45 |
| 0.6 | -0.41 / 6.1e-19 | -0.42 / 3.5e-19 | -0.47 / 1.2e-23 |
| 0.7 | -0.19 / 1.8e-03 | -0.44 / 1.4e-12 | -0.19 / 2.1e-03 |
| 0.8 | +0.24 / 1.9e-02 | -0.44 / 2.4e-05 | -0.09 / 3.9e-01 |
| **Pr.** | **Conv1/2** | **Conv1/3** | **Conv2/3** |
| 0.5 | -0.05 / 1.7e-02 | -0.01 / 6.9e-01 | -0.26 / 3.5e-35 |
| **Pr.** | - | **Conv1/2/3** | - |
| 0.5 | - | -0.14 / 9.4e-11 | - |

coefficient is positive 0.24, which is supposed to be negative; and (2) When pruning multiple layers (two layers or three layers), the correlation also weakens *vs.* pruning one layer at the same layerwise pruning ratio 0.5.

**Anomaly ratio.** Fig. 2 shows that for most cases, oracle pruning selects the pruned weights better than random pruning, but there exists a chance (*e.g.*, Fig. 2 (14) Conv1/3 (0.5)) that oracle pruning is worse than random pruning, where the anomaly ratio is 68.10%, larger than 50%.

**Remarks.** Both taken into consideration, we can conclude that the total sparsity of the model affects the validity of oracle pruning. When the total sparsity is beyond a certain level, oracle pruning does not work anymore. We also have the results (Tab. A4 and Fig. A1 in the Appendix C) of using test loss to measure the final test performance. The above conclusion also holds there.

### 3.3.3 Results with ConvNets on CIFAR and ImageNet-1K

**Kendall correlation.** Fig. 3 shows that on the recent standard convolutional networks, the correlation between pruned train loss and final test accuracy is also pretty weak - the Kendall coefficients are 0.02 (ResNet56 with CIFAR10), 0.01 (VGG19 with CIFAR100), and 0.19 (ResNet18 with ImageNet-1K), respectively. In other words, oracle pruning does not hold in these cases.

**Anomaly ratio.** The anomaly ratio on VGG19 is 71.20%, much higher than 50%. Namely, the oracle pruning idea in this case performs even worse than randomly sampling filters to prune. On ResNet56 and ResNet18, the anomaly ratios are 25.3% and 36.88%, respectively, which suggests oracle pruning is better than random pruning. However, the ratios are still noticeable; if we consider the cost when exhaustively searching the oracle pruning solution, oracle pruning is not a very wise option.

**Remarks.** On modern networks like VGG and ResNet, from the CIFAR datasets level, the correlation between pruned train loss and the final performance becomes very weak, along with noticeable anomaly ratios. Namely, oracle pruning starts to turn invalid on modern convolutional networks, even if the networks (*e.g.*, ResNet56) are not very large. Additionally, results of the test loss in Appendix C also support our statement.

### 3.3.4 Results with ViT-B/16 on ImageNet-1K

Different from convolutional networks, transformers based on the attention mechanism (Vaswani et al., 2017) are a more recent paradigm to build deep vision backbones. The validity check with ViTs is also of interest. Here we prune the heads of ViT-B/16 on ImageNet-1K. Due to the large training cost, we randomly sample 10 pruning combinations for analysis, results presented in Tab. A5.

**Kendall correlation.** As shown in Tab. A5, the correlation between pruned train loss and final test accuracy is completely against our expectation - they should show a negative correlation, but now it is positive.

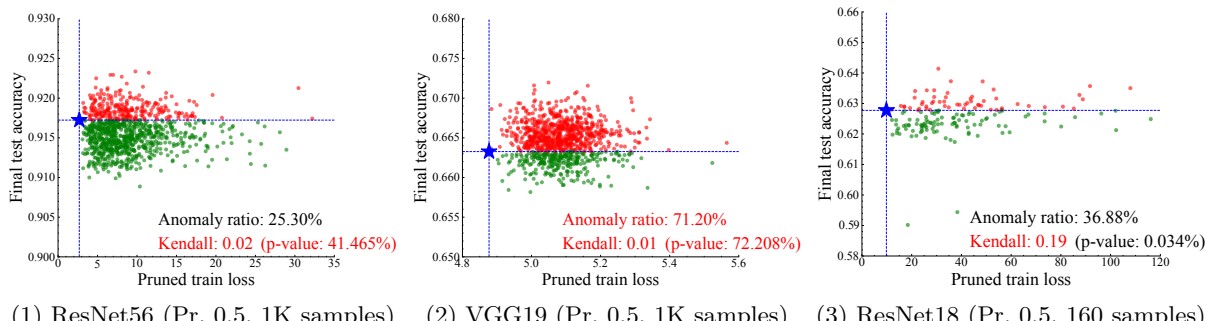

(1) ResNet56 (Pr. 0.5, 1K samples)  (2) VGG19 (Pr. 0.5, 1K samples)  (3) ResNet18 (Pr. 0.5, 160 samples)

Figure 3: Pruned train loss *vs.* final test accuracy with ResNet56, VGG19, and ResNet18. The subcaptions correspond to the pruning settings.

**Counterexample ratio.** Among the 10 random pruning combinations, there are 26 counterexamples[4], accounting for 58% of all 45 comparison pairs.

**Remarks.** The results suggest that on modern attention-based backbones, oracle pruning is also invalid.

Table 3: Pruning results on TinyLLaVA-3.1B. The final test performance is averaged on 5 benchmark datasets following standard practices (see Tab. A6 in the Appendix C for the detailed results).

| Method | #Params | Pruned train loss | Final test performance |
|---|---|---|---|
| Dense model | 3.1B | / | 77.15 |
| ONP | 2B | 1.35 | 76.70 |
| UMP | **2B** | **1.24** | **76.75** |
| GMP | 2B | 1.17 | 76.30 |
| PNP | 2B | 1.16 | 76.28 |

### 3.3.5  Results with TinyLLaVA-3.1B

Finally, we check the validity of oracle pruning using a MLLM: TinyLLaVA-3.1B. The model has 3.1B parameters, which is 36× larger than the last largest model (ViT-B/16) we investigated in this paper. Due to the huge training cost, we can only have 4 pruned models (*e.g.*, UMP, GMP, ONP, and PNP) here, serving for counterexample analysis. The model is evaluated on three image-based question-answering benchmarks: GQA (Hudson & Manning, 2019), ScienceQA-IMG (Lu et al., 2022), and TextVQA (Singh et al., 2019), along with two comprehensive benchmarks: POPE (Li et al., 2023a) and MM-Vet (Yu et al., 2023). The detailed pruning strategies are present in Appendix D.

The results in Tab. 3 show that the validity issue of oracle pruning also applies to the pruning of MLLMs - no strong correlation between the pruned train loss and the final test performance is observed. UMP yields the best test results, but its pruned train loss is worse than GMP and PNP. Additionally, ONP, the second-best pruning approach, shows significantly higher pruned train loss than GMP and PNP, serving as the counterexamples of oracle pruning.

**Conclusive remarks.** On modern networks (after 2012), including representative convolutional networks, ViTs, residual or non-residual networks, and a very recent MLLM, from small datasets (like CIFAR) to large-scale datasets (like ImageNet-1K and the MLLM five evaluation datasets), **all the results suggest oracle pruning does not hold on modern AI models and datasets**.

## 4  What Makes Oracle Pruning Invalid?

The results so far suggest that oracle pruning only holds in the toy case of pruning LeNet5-Mini with small pruning ratios (Tab. 2), but becomes invalid on larger models and datasets. This raises the question: *What*

---

[4]Specific counterexamples: (7, 10), (7, 8), (8, 10), (9, 10), (8, 9), (1, 9), (2, 7), (2, 8), (2, 10), (3, 7), (3, 8), (3, 10), (1, 4), (4, 7), (4, 8), (4, 9), (4, 10), (6, 8), (6, 10), (2, 5), (3, 5), (4, 5), (5, 6), (5, 7), (5, 8), (5, 9)

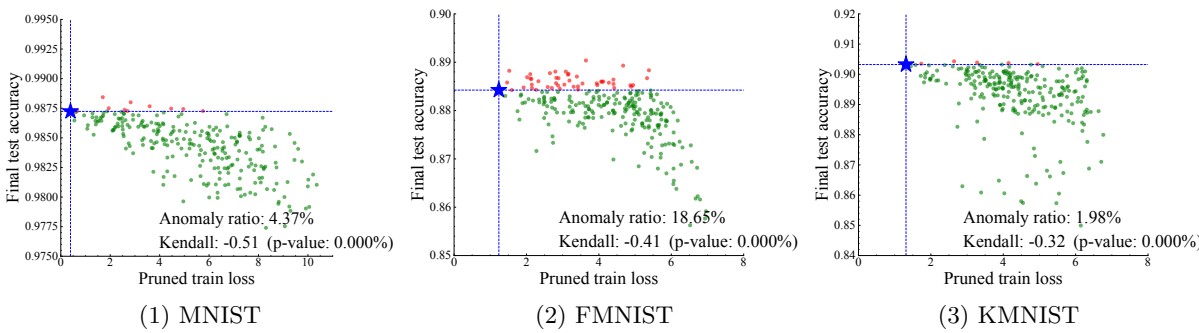

Figure 4: Pruned train loss *vs.* final test accuracy on the variants of MNIST dataset, with LeNet5-Mini network (pruning ratio 0.5, Conv1 layer). FMNIST and KMNIST are two drop-in replacements of MNIST, which are more complex. As seen, the correlation becomes weaker on more challenging datasets.

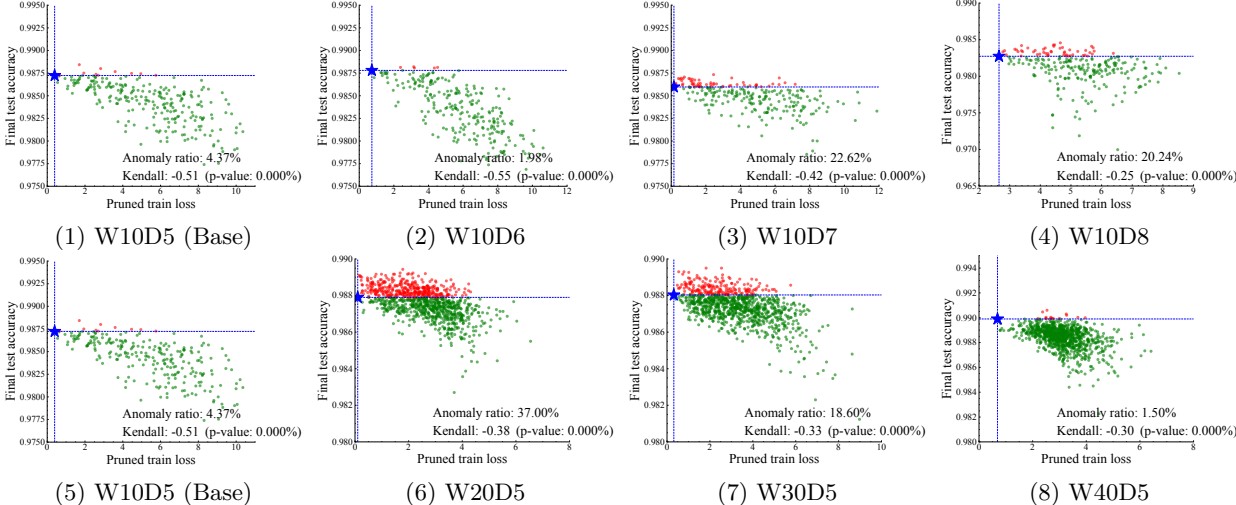

Figure 5: Pruned train loss *vs.* final test accuracy on MNIST with different variants of LeNet5-Mini (pruning ratio 0.5, Conv1 layer). The original LeNet5-Mini (Base) has 5 layers (D5), and each layer has 10 neurons (W10). Here, we change the model width and depth to obtain different variants. As seen, the correlation becomes weaker when pruning more complex networks.

*makes oracle pruning invalid?* The model sparsity is one reason we have identified in Tab. 2. What else? It is quite straightforward to have the hypothesis that the rising task complexity (more specifically, data complexity and model complexity) makes oracle pruning invalid since it is the major change from the LeNet5 era in the 1980s to the recent AI era after 2012.

**Data complexity.** We trained the same model using different datasets with the same pruning scheme. Specifically, we experiment with LeNet5-Mini on MNIST and two of its more complex variants: FMNIST[5] and KMNIST[6] . They are the drop-in replacements of MNIST, but harder.

**Model complexity.** Using LeNet5-Mini as the baseline, we increase the network depth and width while keeping the pruning strategy unchanged. Due to the feature map size limitations after three convolutional layers in LeNet5-Mini, it is not feasible to add more convolutional layers. Therefore, we increase the depth by adding more fully connected layers. For width, since our pruning strategy involves pruning 50% of the filters in only the first convolutional layer, we increase the width by adding more filters only to this first convolutional layer. All networks are trained on the MNIST dataset.

**Experimental results.** Fig. 4, Fig. 5, and Fig. 6 show that the correlation between pruned train loss and final test accuracy turns lower for the models trained on FMNIST and KMNIST *vs.* those trained on MNIST;

---

[5]https://github.com/zalandoresearch/fashion-mnist
[6]https://github.com/rois-codh/kmnist

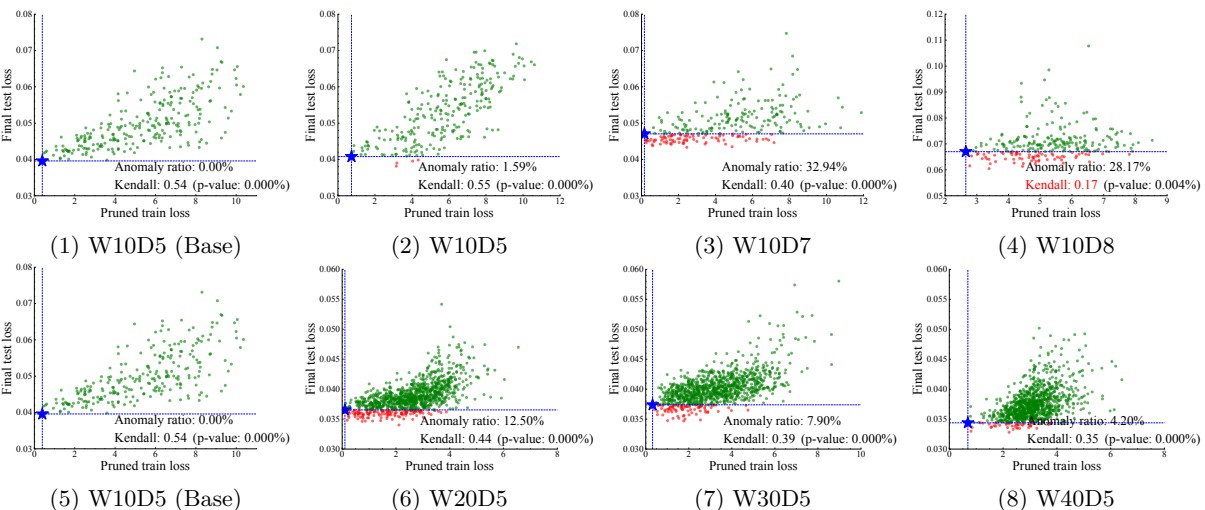

Figure 6: Pruned train loss *vs.* final test loss on MNIST with different variants of LeNet5-Mini (pruning ratio 0.5, Conv1 layer). The original LeNet5-Mini (Base) has 5 layers (D5), and each layer has 10 neurons (W10). Here, we change the model width and depth to obtain different variants. As seen, the correlation becomes *weaker* when pruning *more complex* networks.

the correlation strength also declines with the increased model depth and width. Both pieces of evidence support our hypothesis that the rising task complexity can make oracle pruning invalid.

## 5 Conclusion and Discussion

Oracle pruning has laid the foundation for many pruning criteria in the past three decades. Its validity, however, has not been formally studied for deep models. This work fills the gap by analyzing the correlation between the pruned train loss and the final test performance, along with two other metrics (anomaly ratio and counterexample ratio). Extensive results on a wide range of networks and datasets (from toy networks like LeNet5-Mini to MLLMs; 37K models are trained) suggest a surprising conclusion: **For a practical problem nowadays (starting at a surprisingly basic level such as ResNet56 on CIFAR10), the idea of oracle pruning does *not* hold.** Our further analyses indicate that the increasing task complexity should be considered a key contributing factor. The findings of our work further suggest that it is essential to take into account the retraining process when developing the pruning criterion - only a fraction of retraining is needed to significantly improve the correlation *w.r.t.* the final performance (due to limited length, this part is deferred to the appendix - see the specific results in Appendix E).

Besides, this work has other implications as follows:

(1) This helps explain some mysterious phenomena in network pruning. *E.g.*, the simple magnitude pruning method has long been considered a baseline approach (LeCun et al., 1990), while recently it has been found by several works (Gale et al., 2019; Wang et al., 2023; Renda et al., 2020; Frankle, 2023) comparable or even better than many more advanced pruning criteria derived from oracle pruning. To our best knowledge, few works have systematically addressed this counterintuitive phenomenon[7]. Now with our results, it becomes evident that this phenomenon should not be surprising in the first place, since the underlying assumption of oracle pruning simply does not hold in these cases.

(2) When developing a pruning algorithm, ignoring the subsequent retraining process (if any) is inappropriate. This is not just using the same retraining configurations as suggested by (Blalock et al., 2020; Wang et al., 2023); the retraining process should be accounted for in the design of the pruning algorithm.

(3) Correlation analyses presented in this work are intended to serve as a sanity check to evaluate the effectiveness of any pruning criterion or algorithm.

---

[7]Wang et al. (2023) made one of the few attempts and pointed out that the retraining learning rate has a significant impact.

Notably, while this paper has primarily focused on *retraining-required* pruning methods (*i.e.*, the methods that falls into the conventional three-stage pruning pipeline group - training, pruning, retraining), it is essential to acknowledge that there are situations where oracle pruning remains effective. For non-retraining pruning methods like SparseGPT (Frantar & Alistarh, 2023) and ROSE (Su & Wang, 2026), the Taylor expansion-based form of oracle pruning has been shown to outperform simple magnitude pruning. This suggests that oracle pruning remains valid in specific contexts - particularly when retraining is not involved.

We hope the empirical studies in this work can shed some new light on understanding pruning and help develop more effective pruning algorithms in the long run.

## Acknowledgments

This paper is supported by Young Scientists Fund of the National Natural Science Foundation of China (NSFC) (No. 62506305), Zhejiang Leading Innovative and Entrepreneur Team Introduction Program (No. 2024R01007), Key Research and Development Program of Zhejiang Province (No. 2025C01026), Scientific Research Project of Westlake University (No. WU2025WF003). It is also supported by the research funds of the National Talent Program and Hangzhou Municipal Talent Program.

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

## Appendix

In Appendix A, we provide details of experimental settings, including the usage of networks and datasets, training setting details, and pruning ratio setting details. We conduct more experiments, including extending the analysis framework to other pruning criteria and pruning pipeline in Appendix B. We present more results in Appendix C to support our statement in this paper. We then provide detailed pruning methods for MLLM pruning in Appendix D. We further conduct exploratory experiments in Appendix E. Finally, we discuss future directions in Appendix F.

## A  Experimental Setting Details

### A.1  Details of Networks and Datasets

For the MNIST and CIFAR datasets, we train the original dense model from scratch with accuracies comparable to those in the original papers. For the ImageNet dataset, we use pre-trained models from torchvision (Marcel & Rodriguez, 2010) as the original dense model. For TinyLLaVA-3.1B (Zhou et al., 2024), we employ the pre-trained model on HuggingFace[8] as the base model.

For MLLM pruning experiments, TinyLLaVA-3.1B (Zhou et al., 2024) is a lightweight multimodal language model based on Phi-2 (2.7B) (Li et al., 2023b), a compact variant of LLaMA. It combines vision and language understanding capabilities, is capable of processing both image and text inputs, and is suitable for resource-constrained environments. The model has 3.1B parameters, comprising SigLIP (Zhai et al., 2023), a visual encoder that converts images into feature vectors, and an MLP-based projector that generates text responses. This architecture balances performance and efficiency, making it the chosen base model for our pruning experiments. Additionally, we provide a brief overview of evaluation benchmarks for TinyLLaVA-3.1B as follows.

- **GQA** (Hudson & Manning, 2019) utilizes data organized according to the scene graph structure from the Visual Genome dataset. This benchmark focuses on evaluating a model's proficiency in visual and compositional reasoning.

---

[8] https://huggingface.co/bczhou/TinyLLaVA-3.1B

Table A1: Training setting summary. For the solver, the momentum and weight decay are in brackets. For CIFAR10, batch size 256 is used for retraining instead of 128, which is for saving training time. For LR policy, total epoch, and batch size, the first one is for the pre-training stage, the second is for the retraining stage.

| Network & Data | Solver | LR policy (pre-train and retrain) | Total epoch | Batch size |
|---|---|---|---|---|
| LeNet5-Mini | SGD (0.9, 1e-4) | Multi-step (0:1e-2, 20:1e-3) | 30 | 256 |
| (MNIST) | | Multi-step (0:1e-3, 20:1e-4) | 30 | 256 |
| ResNet56 | SGD (0.9, 5e-4) | Multi-step (0:1e-1, 100:1e-2, 150:1e-3) | 200 | 128 |
| (CIFAR10) | | Multi-step (0:1e-2, 60:1e-3, 90:1e-4) | 120 | 256 |
| VGG19 | SGD (0.9, 5e-4) | Multi-step (0:1e-1, 100:1e-2, 150:1e-3) | 200 | 256 |
| (CIFAR100) | | Multi-step (0:1e-2, 60:1e-3, 90:1e-4) | 120 | 256 |
| ResNet18 | SGD (0.9, 1e-4) | - | - | - |
| (ImageNet) | | Multi-step (0:1e-2, 10:1e-3, 20:1e-4) | 30 | 256 |
| ViT-B/16 | Adam (0.9, 3e-1) | - | - | - |
| (ImageNet) | | Cosine (1.5e-4) | 300 | 1024 |
| TinyLLaVA-3.1B | Adam (0.9, 3e-1) | - | - | - |
| (LLaVA-1.5 Dataset) | | - | 2 | 8 |

- **TextVQA** (Singh et al., 2019) involves a dataset of image-question pairs, with text incorporated into the images. It tests the model's ability to not only recognize textual information but also perform reasoning based on the text.

- **ScienceQA-IMG** (Lu et al., 2022) is a subset of the ScienceQA benchmark, which consists of scientific questions along with relevant contexts. The evaluation centers on a model's capacity for reasoning in scientific domains by predicting correct answers from the given context.

- **POPE** (Li et al., 2023a) is a benchmark aimed at assessing hallucination issues in MLLMs. By using samples of both positive and negative objects, it effectively evaluates whether models can correctly identify real samples while avoiding recognition of non-existent entities, thereby measuring hallucination tendencies.

- **MM-Vet** (Yu et al., 2023) provides a comprehensive assessment of LMMs across complex multimodal tasks. Using GPT-4 as an evaluator, MM-Vet examines six dimensions of LMM performance, including visual recognition, spatial reasoning, common knowledge inference, language generation, visual math, and OCR capabilities.

## A.2 Details of Training Setting

Regarding the evaluation architecture, we intentionally use ResNet instead of AlexNet and VGG on ImageNet because the single-branch architecture is no longer representative of modern deep network architectures with residuals, but we still retain VGG19 on the CIFAR analysis to ensure that statements are not limited to a specific architecture. At the same time, we also use ViT-B/16 on ImageNet to increase the diversity of evaluation architectures. In addition to the key settings mentioned in the paper, a more detailed summary of the training settings is provided in Tab. A1.

To ensure the stability of the pruning results, we repeated three times for each combination. For ResNet18, ViT-B/16, and TinyLLaVA-3.1B, we only repeat each pruning combination once due to the training cost.

## A.3 Details of Pruning Ratios

Due to the limitation of computing resources and training time, we only conducted a full pruning ratio specific study on MNIST with LeNet5-Mini. For the rest, we use a single standard pruning ratio strategy.

Before we list the specific pruning ratios, we explain how we set them:

(1) For LeNet5-Mini, there are three conv layers that can be pruned, we will use a list of 3 floats to represent its pruning ratios for the 3 conv layers. For example, "[0.5, 0, 0.5]" means "for the second conv layer, the pruning ratio is 0; the other two conv layers have a pruning ratio of 0.5".

Table A2: Pruning ratio summary. See more detailed statements in Appendix A.3.

| Dataset | Network | Pruning ratio |
|---|---|---|
| MNIST | LeNet5-Mini | [0.2-0.8, 0, 0] |
| | | [0, 0.2-0.8, 0] |
| | | [0, 0, 0.2-0.8] |
| | | [0, 0.5, 0.5] |
| | | [0.5, 0, 0.5] |
| | | [0.5, 0.5, 0] |
| | | [0.5, 0.5, 0.5] |
| CIFAR10 | ResNet56 | [0, 0.5, 0.5, 0.5] |
| CIFAR100 | VGG19 | [0-15:0.5] |
| ImageNet | ResNet18 | [0, 0.5, 0.5, 0.5, 0.5] |
| ImageNet | ViT-B/16 | [0-11: 0.5] |
| LLaVA-1.5 Dataset | TinyLLaVA-3.1B | 0.4375 |

(2) For a ResNet, if it has N stages, we will use a list of N floats to represent its pruning ratios for the N stages. For example, ResNet56 has 4 stages in conv layers, then "[0, 0.5, 0.5, 0.5]" means "for the first stage (which is also the first conv layer), the pruning ratio is 0; the other three stages have a pruning ratio of 0.5". Besides, since we do not prune the last conv layer in a residual block, which means for a two-layer residual block, we only prune the first layer.

(3) For VGG19, we apply the following pruning ratio setting. For example, "[0-15:0.5]" means "for conv layer 0 to 15, the pruning ratio is 0.5".

(4) For a ViT, we prune the attention heads and feedforward neural network (FNN) in all encoder layers. For example, ViT-B/16 has 12 encoder layers, then "[0-11: 0.5]" stands for "for layer 0 to 11, the pruning ratio is 0.5"

(5) For a TinyLLaVA-3.1B, we apply unstructured pruning to the LLM component with a pruning rate of 0.4375 for different strategies. The vision encoder and projector components remain unpruned, accounting for 14.5% of the total model.

Accordingly, the detailed pruning ratios used in each experiment are presented in Tab. A2.

# B  Supplementary Experiments

## B.1  Extending The Analysis Framework to Other Pruning Criteria

We further apply our analysis framework to examine other pruning criteria. Specifically, we select two popular pruning methods (*e.g.*, magnitude pruning and Taylor-FO pruning (Molchanov et al., 2017)). For each method, we compute the magnitude of pruned parameters and the Taylor-FO score of pruned parameters as counterparts to the pruned train loss and final test performance in our framework to calculate the correlation. The experimental settings and analysis are presented as follows.

**Experimental Setting.** We use ResNet50 as the base model and CIFAR100 as the dataset. For the pruning setup, we prune the Conv1 and Conv2 layers of each block with a pruning ratio of 50%, following a standard pruning configuration. During retraining, we adopt SGD (momentum 0.9, weight decay 1e-4) as the optimizer, with a multi-step learning rate schedule (0–59 epochs: 1e-2, 60–89: 1e-3, 90–119: 1e-4). The total training lasts for 120 epochs with a batch size of 256. We randomly select a subset of CIFAR100 (20k samples) as calibration data for computing the Taylor-FO score.

**Experimental Results.** For magnitude pruning, the Kendall correlation between the magnitude of pruned parameters and the final test accuracy is -0.006 (p-value = 0.91), and the correlation with the final test loss is 0.038 (p-value = 0.47). These results indicate a negative correlation with accuracy and a positive correlation with loss, consistent with the intuition of magnitude pruning (i.e., removing parameters with smaller magnitudes tends to preserve model performance). In contrast, for Taylor-FO pruning, we observe a

positive correlation between the Taylor-FO score of pruned parameters and the final test accuracy (0.046, p-value = 0.39) and a negative correlation with the final test loss (-0.009, p-value = 0.86). This observation contradicts the underlying idea of Taylor-FO pruning, where parameters with higher Taylor-FO scores are considered more important, and pruning them should degrade performance. Overall, these results suggest that while magnitude pruning remains consistent with its expected behavior, Taylor-FO pruning, as an extension of the Oracle Pruning idea, fails to show validity when applied to post-training pruning.

Table A3: Kendall correlation between pruned train loss at each iterative pruning step and final test performance on ResNet50 with CIFAR100. Each entry in the table is arranged as coefficient / p-value.

| Iteration | Pruned Train Loss vs Final Test Loss | Pruned Train Loss vs Final Test Acc. |
|---|---|---|
| **LR1 (first four iterations: 0.001; final stage: 0:0.1, 20:0.01, 40:0.001)** | | |
| 1/5 | 0.05 / 0.31 | 0.03 / 0.58 |
| 2/5 | -0.05 / 0.32 | 0.04 / 0.47 |
| 3/5 | 0.02 / 0.71 | 0.04 / 0.41 |
| 4/5 | 0.10 / 0.06 | 0.08 / 0.13 |
| 5/5 | 0.05 / 0.34 | 0.03 / 0.52 |
| **LR2 (first four iterations: 0:0.1, 5:0.001, 10:0.0001; final stage: 0:0.1, 20:0.01, 40:0.001)** | | |
| 1/5 | -0.04 / 0.48 | -0.03 / 0.64 |
| 2/5 | -0.06 / 0.28 | 0.003 / 0.95 |
| 3/5 | -0.12 / 0.02 | 0.03 / 0.55 |
| 4/5 | -0.08 / 0.14 | 0.07 / 0.19 |
| 5/5 | 0.02 / 0.72 | 0.07 / 0.22 |

### B.2 Extending The Analysis Framework to Other Pruning Pipeline

To further validate the generality of our analytical framework beyond the one-shot pruning setting used in the main paper, we additionally extend our correlation study to iterative pruning pipelines. Specifically, we evaluate whether intermediate pruning states (after each pruning round) provide predictive signals regarding the final restored performance. This allows us to examine whether the observation that "pruned train loss does not correlate with final test performance" continues to hold even under pruning-retraining cycles, where more optimization signals are exposed throughout the pruning trajectory.

**Experimental Setting.** We perform experiments on ResNet50 with CIFAR100 under a 5-step iterative pruning scheme. Each round removes 10% of parameters (50% total), followed by retraining: (1) The first four rounds are retrained for 15 epochs each, and (2) The final round is retrained for 60 epochs. This yields 120 total retraining epochs (*i.e.*, identical to the epoch setting in the main paper). After every pruning step, we record the train loss of the newly pruned model, then compute the Kendall correlation between pruned train loss and final test performance. We test two learning rate schedules (LR1 and LR2), enabling a more comprehensive evaluation of robustness across optimization settings.

**Experimental Results.** Across both LR schedules, we observe no significant positive correlation between pruned train loss and final test loss, nor significant negative correlation between pruned train loss and final test accuracy (as shown in Table A3). This implies that even with access to intermediate checkpoints, the iterative pruning trajectory still does not present reliable oracle predictability regarding eventual recovery, which further supports our conclusion that pruned loss is not a performance oracle under iterative pruning.

## C Supplementary Results

**Results with LeNet5-Mini on MNIST.** We add some pruning results on MNIST, providing the results of pruned train loss *vs.* final test loss, as shown in Tab. A4 and Fig. A1 (Fig. A2 and A3 are supplementary results with more pruning ratios).

Table A4: Kendall correlation between pruned train loss and final test loss, by exhaustively pruning LeNet5-Mini network on MNIST dataset. Each entry in the table is arranged as Kendall coefficient / p-value. *Pr.* means the pruning ratio for the corresponding layer combination of Conv1, Conv2, and Conv3. The red entries mean these results pose weak or counterintuitive correlations.

| Pr. | Conv1 | Conv2 | Conv3 |
|---|---|---|---|
| 0.2 | 0.57 / 3.1e-08 | 0.74 / 6.0e-13 | 0.66 / 1.4e-10 |
| 0.3 | 0.53 / 7.3e-18 | 0.67 / 2.9e-27 | 0.55 / 3.5e-19 |
| 0.4 | 0.48 / 1.2e-24 | 0.65 / 1.8e-44 | 0.54 / 7.3e-32 |
| 0.5 | 0.54 / 2.6e-37 | 0.56 / 1.5e-40 | 0.57 / 4.8e-42 |
| 0.6 | 0.45 / 2.8e-22 | 0.44 / 5.5e-21 | 0.46 / 3.5e-23 |
| 0.7 | 0.20 / 1.0e-03 | 0.45 / 2.4e-13 | 0.20 / 1.1e-03 |
| 0.8 | -0.25 / 1.4e-02 | 0.41 / 7.1e-05 | 0.10 / 3.3e-01 |
| **Pr.** | **Conv1/2** | **Conv1/3** | **Conv2/3** |
| 0.5 | 0.07 / 1.9e-03 | 0.01 / 6.8e-01 | 0.26 / 8.5e-36 |
| **Pr.** | **Conv1/2/3** | | |
| 0.5 | 0.13 / 4.2e-10 | | |

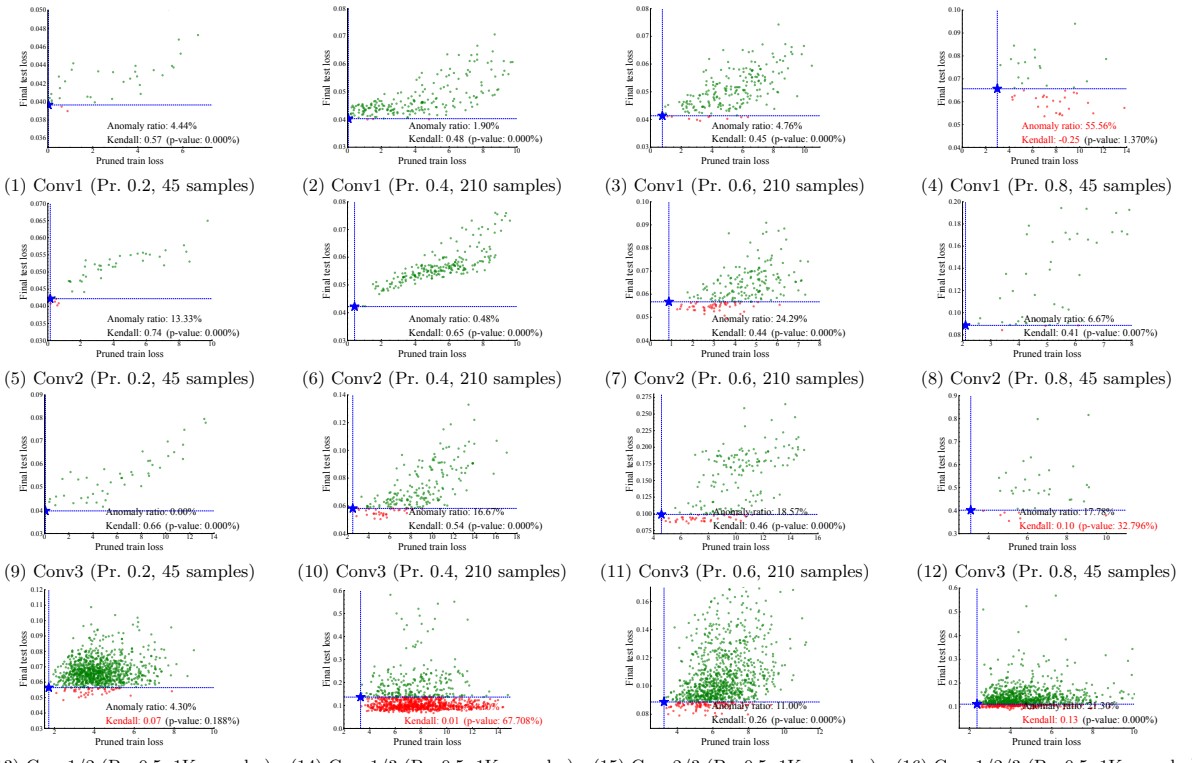

(1) Conv1 (Pr. 0.2, 45 samples)  (2) Conv1 (Pr. 0.4, 210 samples)  (3) Conv1 (Pr. 0.6, 210 samples)  (4) Conv1 (Pr. 0.8, 45 samples)

(5) Conv2 (Pr. 0.2, 45 samples)  (6) Conv2 (Pr. 0.4, 210 samples)  (7) Conv2 (Pr. 0.6, 210 samples)  (8) Conv2 (Pr. 0.8, 45 samples)

(9) Conv3 (Pr. 0.2, 45 samples)  (10) Conv3 (Pr. 0.4, 210 samples)  (11) Conv3 (Pr. 0.6, 210 samples)  (12) Conv3 (Pr. 0.8, 45 samples)

(13) Conv1/2 (Pr. 0.5, 1K samples)  (14) Conv1/3 (Pr. 0.5, 1K samples)  (15) Conv2/3 (Pr. 0.5, 1K samples)  (16) Conv1/2/3 (Pr. 0.5, 1K samples)

Figure A1: Pruned train loss *vs.* final test loss on MNIST with LeNet5-Mini. The subcaptions correspond to the pruning rates of each image. The star denotes the oracle pruning results, where points with final test loss lower than the oracle pruning are marked in red, and those lower are marked in green.

**Results on CIFAR and ImageNet-1K.** We provide some pruning results on CIFAR10/100 (pruned train loss *vs.* final test loss), as shown in Fig. A4. We further report the results of ViT-B/16 on ImageNet-1K in Tab. A5.

**Results with TinyLLaVA-3.1B.** We provide detailed results in Tab. A6.

**Results for Sec. 4.** We provide more results in Fig. A5.

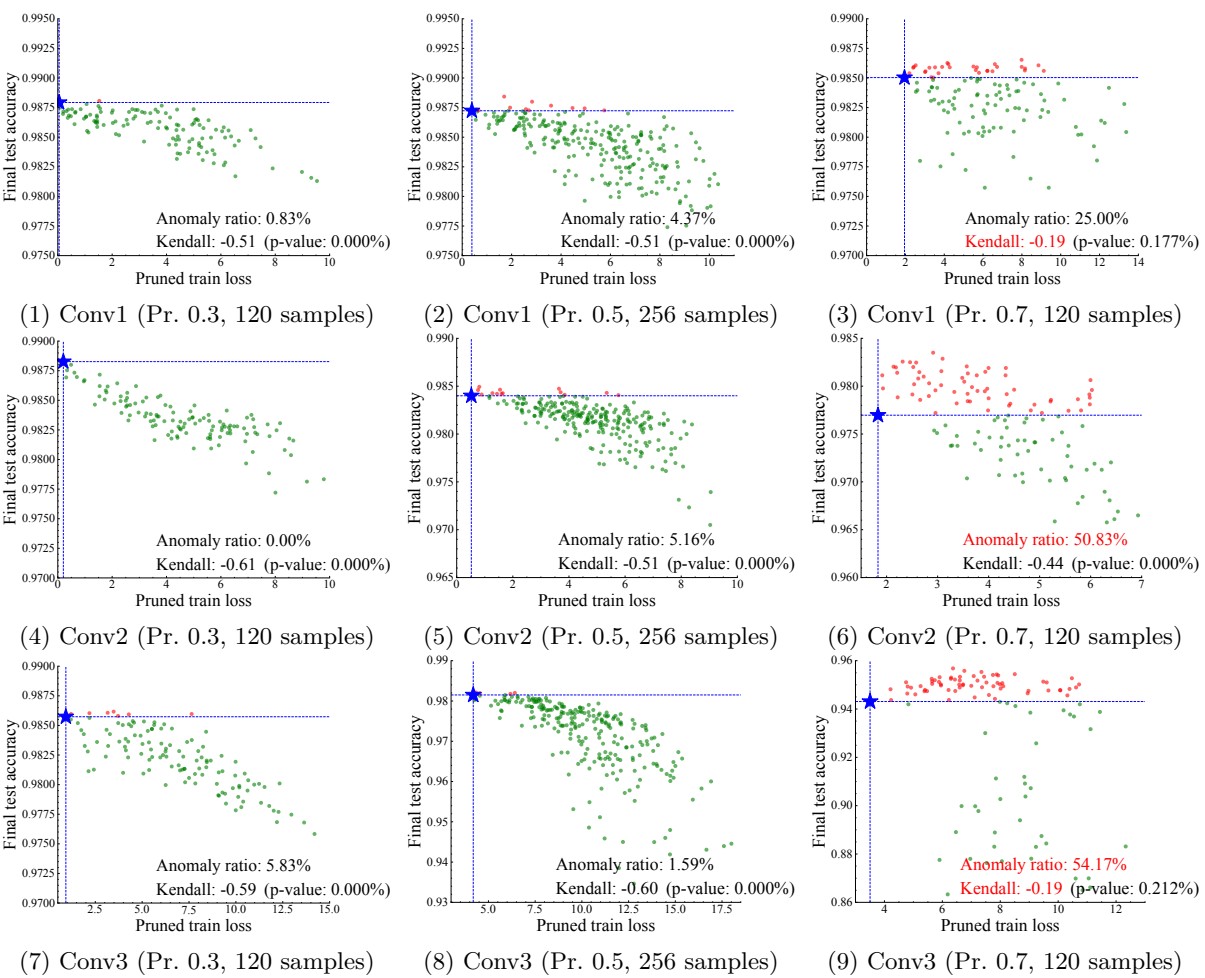

Figure A2: Pruned train loss *vs.* final test accuracy on MNIST with LeNet5-Mini.

Table A5: Results of ViT-B/16 on ImageNet-1K.

| Combination | Pruned train loss | Final test accuracy | Final test loss |
|---|---|---|---|
| 1 | 6.9768 | 76.2445 | 1.9714 |
| 2 | 7.0515 | 75.9731 | 1.9781 |
| 3 | 7.0442 | 76.0662 | 1.9826 |
| 4 | 7.0162 | 76.3223 | 1.9689 |
| 5 | 6.9989 | 74.0748 | 2.0530 |
| 6 | 7.0709 | 75.7066 | 1.9901 |
| 7 | 7.0106 | 75.9135 | 1.9836 |
| 8 | 6.9991 | 75.6824 | 1.9826 |
| 9 | 7.0092 | 76.2862 | 1.9725 |
| 10 | 6.9962 | 74.9903 | 2.0148 |
| Kendall | / | 0.16 (60%) | -0.04 (86%) |

# D  Details of Pruning Strategies in MLLM Pruning

We summarize the unstructured pruning strategies used for MLLMs in this paper as follows.

- **Uniform magnitude pruning (UMP).** The core idea of this method is to calculate the magnitude of each weight in the fully connected layers as the pruning criterion and remove weights with smaller magnitudes (setting them to zero) to reduce computational load. This fundamental and widely used

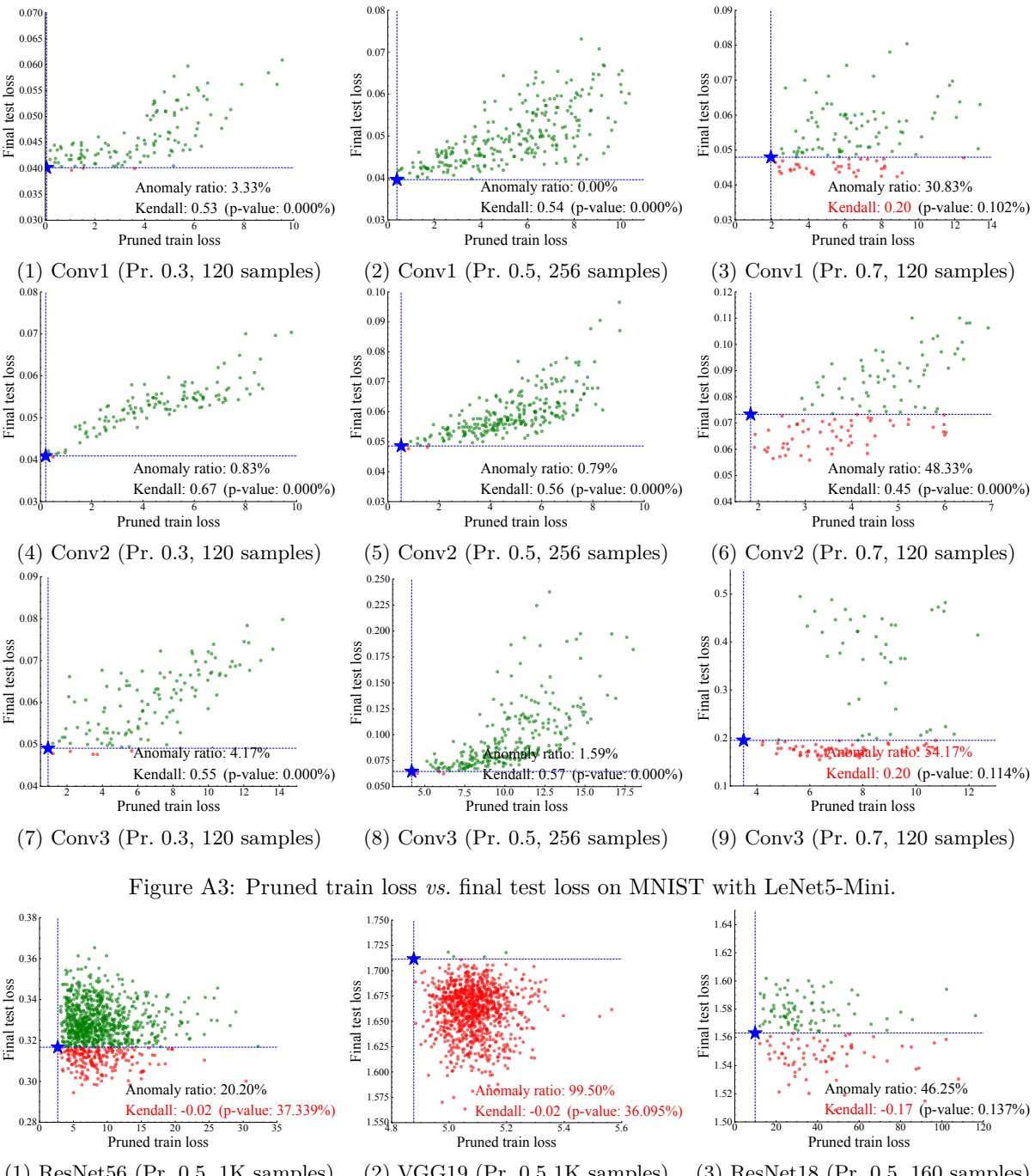

Figure A3: Pruned train loss *vs.* final test loss on MNIST with LeNet5-Mini.

Figure A4: Pruned train loss *vs.* final test loss with ResNet56 (on CIFAR10), VGG19 (on CIFAR100), ResNet18 (on ImageNet-1K).

unstructured pruning strategy, also known as uniform pruning, maintains the same proportion of parameters across all fully connected layers.

- **Global magnitude pruning (GMP).** This method performs unstructured pruning on a global scale; it is not limited to a single layer or specific network structure and unifies the entire neural network for pruning using amplitude sorting. This approach targets weights with smaller magnitudes across the entire network, reducing redundancy.

Table A6: For the pruning results of MLLMs, the table presents the performance of the Base model and the models pruned using the four strategies. We present the model's performance across five benchmarks. *PTL* stands for the pruned train loss.

| Methods | #Params | Vision-Encoder | Res. | PTL | SQA | TextVQA | GQA | MM-Vet | POPE | Avg. |
|---------|---------|---------------|------|-----|-----|---------|-----|--------|------|------|
| Dense model | 3.1B | SigLIP-0.4B | 384 | / | 69.1 | 59.1 | 62 | 32 | 86.4 | 77.15 |
| **UMP** | **2B** | **SigLIP-0.4B** | **384** | **1.24** | **69.4** | **56.7** | **60.1** | **34.2** | **86.6** | **76.75** |
| GMP | 2B | SigLIP-0.4B | 384 | 1.17 | 69.9 | 55.8 | 60.1 | 33 | 86.4 | 76.30 |
| ONP | 2B | SigLIP-0.4B | 384 | 1.35 | 69.8 | 55.7 | 61.5 | 33 | 86.8 | 76.70 |
| PNP | 2B | SigLIP-0.4B | 384 | 1.16 | 69.5 | 55.1 | 61.5 | 33.1 | 86 | 76.28 |

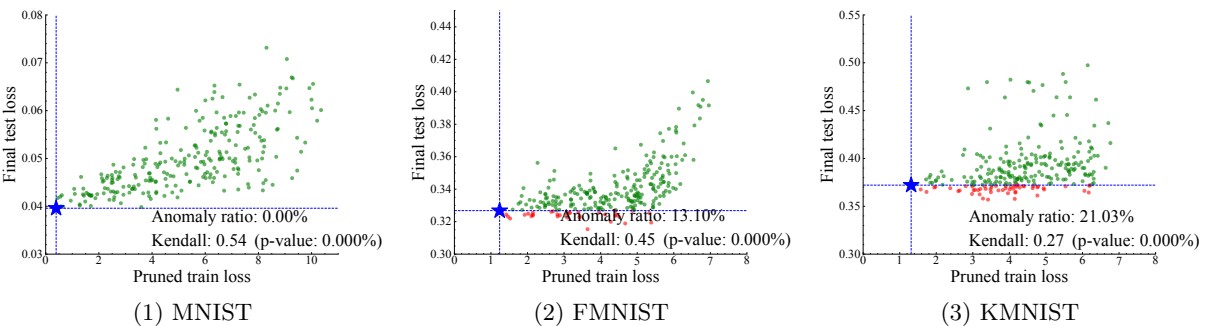

Figure A5: Pruned train loss *vs.* final test loss on the variants of MNIST dataset, with LeNet5-Mini network (pruning ratio 0.5, Conv1 layer). FMNIST and KMNIST are two drop-in replacements of MNIST, which are more complex. As seen, the correlation becomes *weaker* on *more challenging* datasets. See more discussions in Sec. 4.

- **Outlier-based non-uniform pruning (ONP).** Inspired by (Yin et al., 2023), this method begins by evaluating the proportion of outliers in the weights of each layer. Layers with more outliers are regarded as more important and thus assigned a smaller pruning rate, while layers with fewer outliers are assigned a higher pruning rate.

- **PCA-based non-uniform pruning (PNP).** This method seeks to identify low-importance components within each layer's weight matrix by analyzing the main direction of feature extraction via PCA. It then assigns pruning rates linearly based on the proportion of principal components in each layer, so that layers with a higher proportion of principal components receive a lower pruning rate, while those with a lower proportion receive a higher pruning rate.

# E  A Lesson: Retraining Must be Considered in Pruning

Pruning is widely used to improve model efficiency. Now that oracle pruning turns out invalid in giving us a good pruning criterion, what should we pursue towards a better pruning algorithm? Since the results suggest that the model performance before retraining is barely correlated with the performance after retraining, an obvious lesson is that the retraining process must be considered when developing the pruning criterion. Here we present preliminary results to support this argument.

Specifically, we do not assess the pruned models right after pruning. Instead, we retrain them for a short period (only 10% of the original retraining process with a proportionally scaled learning rate schedule) and then assess the model performance by different pruning schemes.

The results in Fig. A6 show that the model performance after full retraining is highly correlated with the performance with only 10% retraining. This implies, in practice, we can assess the pruned model after a short period of retraining (*e.g.*, 10% retraining here) and it will dramatically improve the correlation with the final performance. Future works in pruning may seek more efficient ways to reduce the retraining cost for evaluating different pruning schemes.

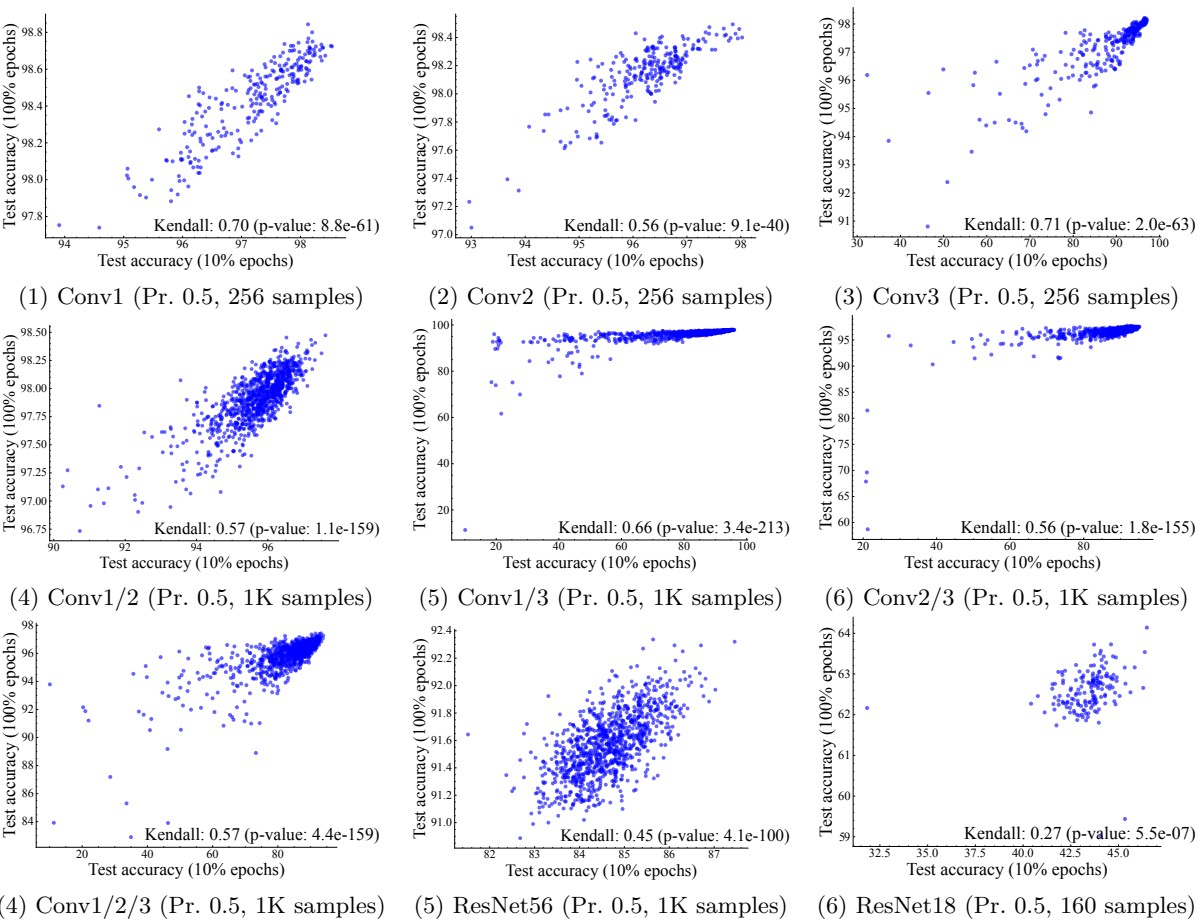

Figure A6: Test accuracy (10% epochs) *vs.* test accuracy (100% epochs).

## F Future Work

With the rapid emergence of new model architectures, pruning has further expanded its application scope to reasoning models (Feng et al., 2025a;c;b; Zhang et al., 2025; Du et al., 2026), diffusion language models (Feng et al., 2026b;a), and omni models (Shao et al., 2025; Wang et al., 2026; Tao et al., 2026; Jin et al., 2025; Ai & He, 2026). Looking forward, how to better apply pruning techniques to these emerging models for building efficient models remains a promising direction.

