# OpenReview forum: "Is Oracle Pruning the True Oracle? -- A Sanity-Check of Neural Network Pruning with Retraining"
_TMLR — Accepted by TMLR_

### Review · Reviewer_G7Rz · 2025-11-08

**Summary Of Contributions:**

This paper conducts extensive results to verify if the oral pruning is actually valid or not, based on the correlation between the dense models and the pruned models after retraining. Across 37K trained models and a broad spectrum of architectures (from LeNet5 and ResNets to ViT and TinyLLaVA-3.1B), the authors find that pruned training loss is weakly correlated—or even uncorrelated—with retrained test performance. They conclude that oracle pruning, a foundation of many pruning criteria, no longer holds for modern models and datasets, and they advocate for incorporating retraining effects into pruning criterion design.

**Additional Comments:**

1. While there are some footnotes in the paper to explain some details, it would be very helpful for readers to understand more details if we could summarize the hypermeter used in the pruning setting in the stage of the paper.

**Audience:**

Yes

**Audience Explanation:**

I personally appreciate papers that aim to fundamentally understand the behavior of algorithms, and this work is a good example. It provides an in-depth investigation into the behavior of oracle pruning, a long-standing assumption in model compression. I believe many researchers in the model compression community will find the findings of this paper highly interesting and valuable.

**Broader Impact Concerns:**

Since pruning is a cornerstone of model compression and energy-efficient AI, understanding when and why certain pruning strategies fail has direct implications for sustainable computing. The findings encourage more principled, empirically grounded compression strategies that truly reduce energy consumption without sacrificing performance.

**Claims And Evidence:**

Yes

**Claims Explanation:**

1. Extensive experiments: The study spans a wide range of models, datasets, and pruning ratios, with 37K runs is impressive for a methodological paper.

2. Clear methodology: The use of Kendall correlation, anomaly ratio, and counterexample ratio provides transparent, quantitative metrics.

3. The paper studied with multiple, common architectures, and datasets.

**Requested Changes:**

1. While I understand the main target of this paper is Oracle pruning., it will be interesting to see if similar results will be observed for other pruning methods like the magnitude pruning. I am happy to see some results with the Resnet50 on Imagenet or cifar100.

2. It is not very clear for me what are measured to calculate the correlation. Are we calculating the correlation between one-hot vectors of two models? Or only the binary results, for example, correct or incorrect, are considered? I will encourage the authors to report both results.

3. There are some excellent papers showing that when we do iterative pruning and retraining, the pruned models will still be linear connected with the dense model. And this will hold for relatively large prune ratio (https://arxiv.org/abs/2210.13738). Intuitively, I believe if the model are still completely connected, they will have a stronger correlation. It would be interesting to verify this as well.

---

> ### Author Response · Authors · 2025-12-23
> **Response to Reviewer G7Rz (Part-1)**
>
> We sincerely thank the reviewer for the positive feedback and constructive comments. We have incorporated the suggestions and revised the manuscript accordingly, with the updated contents highlighted in **blue**.
>
> ---
>
> > **Q1:** *The advice on extending the analysis to other pruning methods.*
>
> **A:** We sincerely thank the reviewer for the insightful advice. Our analysis framework is also applicable to other pruning methods using specific pruning criteria (*e.g.*, magnitude pruning, Taylor-FO pruning[`R1`]). Following the reviewer’s advice, we conducted experiments on ResNet50 using CIFAR100. We adopted standard and fair experimental settings (see the detailed experimental setting in Appendix `B.1`) and report the results as follows.
>
> For magnitude pruning, we observe a negative correlation with the final test accuracy (-0.006, p = 0.91) and a positive correlation with the final test loss (0.038, p = 0.47). This aligns with the intuition of magnitude pruning, where removing low-magnitude parameters typically has minimal impact on performance. In contrast, for Taylor-FO pruning, the correlation with accuracy is positive (0.046, p = 0.39) and with loss is negative (-0.009, p = 0.86), which contradicts its theoretical premise that parameters with higher Taylor-FO scores are more important, and that pruning them should therefore degrade performance. Overall, these findings suggest that while magnitude pruning behaves consistently with its intended design, Taylor-FO pruning, as an extension of the Oracle Pruning idea, does not exhibit validity under post-training pruning.
>
> We appreciate the insightful suggestion by the reviewer and have updated the manuscript accordingly (see more details in Appendix `B.1`).
>
> ---
>
> > **Q2:** *The suggestion of clarifying what is measured in the correlation calculation and additional results*
>
> **A:** We sincerely thank the reviewer for the helpful suggestion. When calculating the correlation (refer to Figure 1), we calculate the correlation between the **pruned train loss** (*i.e.*, the loss of pruned model on training set) and the **final test performance** (*i.e.*, test accuracy or test loss of retrained model). We reported both correlation results calculated using accuracy and loss (*e.g.*, Table `2` / `A3`). We appreciate this thoughtful suggestion, and we have updated the manuscript to highlight the metrics we used for correlation calculation (refer to *“Correlation between what?”* in Section `3.1`).

---

> ### Author Response · Authors · 2025-12-23
> **Response to Reviewer G7Rz (Part-2)**
>
> > **Q3:** *The advice about verifying the conclusion on iterative pruning*
>
> **A:** We sincerely thank the reviewer for the constructive feedback. We would like to first clarify—gently and precisely—that the “linearly connected” path mentioned in the review refers to the relationship between the original dense model and the iteratively pruned–retrained models (corresponding to [`R2`, `R3`]). This differs from our setting, where we study the relationship between a pruned model and its subsequent retrained model (*i.e.*, pruned model <-> final model).
>
> To address the reviewer’s concern, we further conducted experiments on ResNet-50 with CIFAR-100 under an iterative pruning regime. We applied 5 pruning rounds, each removing 10% of the parameters (50% in total). The first four rounds were followed by 15 epochs of retraining, and the final round by 60 epochs, yielding 120 retraining epochs in total—identical to the one-shot pruning schedule used in the main paper. Iterative pruning provides intermediate pruned checkpoints, allowing us to record the train loss after each pruning step. We then computed the correlation between the pruned train losses and the final test performance under two different learning-rate schedules. Across both settings, we consistently observed no significant positive correlation between pruned train loss and final test loss, and no significant negative correlation between pruned train loss and final test accuracy. These findings further support our claim under the iterative pruning setting.
>
> *LR1: (first four iterations: lr = 0.001; final iteration: 60 epochs with lr schedule (0: 0.1, 20: 0.01, 40: 0.001))*
> | Iteration | Pruned Train Loss vs Final Test Loss (τ, p) | Pruned Train Loss vs Final Test Acc (τ, p) |
> |-|-|-|
> | 1/5 | (0.05, 0.31)  | (0.03, 0.58) |
> | 2/5 | (-0.05, 0.32) | (0.04, 0.47) |
> | 3/5 | (0.02, 0.71)  | (0.04, 0.41) |
> | 4/5 | (0.10, 0.06)  | (0.08, 0.13) |
> | 5/5 | (0.05, 0.34)  | (0.03, 0.52) |
>
> *LR2: (first four iterations: 15 epochs with lr schedule (0:0.1, 5:0.001, 10:0.0001); final iteration: 60 epochs with lr schedule (0: 0.1, 20: 0.01, 40: 0.001))*
> | Iteration | Pruned Train Loss vs Final Test Loss (τ, p) | Pruned Train Loss vs Final Test Acc (τ, p) |
> |-|-|-|
> | 1/5 | (-0.04, 0.48) | (-0.03, 0.64) |
> | 2/5 | (-0.06, 0.28) | (0.003, 0.95) |
> | 3/5 | (-0.12, 0.02) | (0.03, 0.55)  |
> | 4/5 | (-0.08, 0.14) | (0.07, 0.19)  |
> | 5/5 | (0.02, 0.72)  | (0.07, 0.22)  |
>
> We thank the reviewer for the valuable suggestion, and have incorporated corresponding updates into Appendix `B.2` of the updated manuscript.
>
> ---
>
> > **Q4:** *The additional comments about providing the hyperparameters in the pruning setting.*
>
> **A:** We sincerely thank the reviewer for the helpful advice. We agree that providing hyperparameter settings can help readers better understand the experimental details. For the post-training pruning discussed in this paper, the main pruning hyperparameters include the pruning target (*i.e.*, specific layers or architectures to prune) and the pruning ratio. We have detailed the pruning targets and ratios for all experiments (*e.g.*, from LeNet5-Mini to TinyLLaVA-3.1B) in Appendix `A.3` and summarized them in Table `A2` of the manuscript. We thank the reviewer again for this kind reminder, and we further highlight the related statements in the updated manuscript (see more in Section `3.3.1` & Appendix `A.3`).
>
> ---
>
> **References:**
>
> - [`R1`] Molchanov, Pavlo, et al. "Pruning convolutional neural networks for resource efficient inference." arXiv preprint arXiv:1611.06440 (2016).
> - [`R2`] Jin, Tian, et al. "Pruning’s effect on generalization through the lens of training and regularization." Advances in Neural Information Processing Systems 35 (2022): 37947-37961.
> - [`R3`] Frankle, Jonathan, et al. "Linear mode connectivity and the lottery ticket hypothesis." International Conference on Machine Learning. PMLR, 2020.
>
> ---
>
> Last but not least, we would like to sincerely thank Reviewer `G7Rz` again for the valuable time and constructive feedback provided during this review.
>
> **Thank you for helping improve our work so far! We are actively available during the discussion period. Let us know should you have any further questions.**

---

> > ### Comment · Reviewer_G7Rz · 2026-05-01
> > **Thanks for addressing my comments**
> >
> > I thank the authors for the comments. My concerns are addressed.

---

> > > ### Author Response · Authors · 2026-05-01
> > > **Thanks**
> > >
> > > Dear Reviewer `G7Rz`,
> > >
> > > Thank you very much for revisiting our work. We truly appreciate your recognition of our study, which is very encouraging to us. We are also grateful for the time and care you devoted to reading our rebuttal and for your constructive feedback throughout the process.
> > >
> > > Best regards,
> > >
> > > The Authors of Submission 6219

---

### Review · Reviewer_cYmH · 2025-12-18

**Summary Of Contributions:**

**The contributions of the paper**:

This paper examines the validity of oracle pruning for modern deep neural networks, which removes unimportant weights based on minimizing the pruned training loss. Concretely, the authors analyze the correlation between model performance before and after retraining within the three-stage pruning pipeline (pre-training, pruning, and retraining) through extensive experiments (37K models are trained) across a wide spectrum of models (LeNet5, VGG, ResNets, ViT, MLLM) and datasets (MNIST and its variants, CIFAR10/CIFAR100, ImageNet-1K, MLLM data).

**Strengths**:

The authors conduct extensive experiments (37K models are trained) to analyze the correlation between model performance before and after retraining within the three-stage pruning pipeline (pre-training, pruning, and retraining).

**Weaknesses**:

1. In Section 1, the introduction to oracle pruning could be improved for clarity. Specifically, the notation and explanation for Equations (1) and (2) are currently confusing. Moreover, I cannot see the necessity of splitting the hessian term into two parts; it may help to follow the exposition style used in related literature. Finally, the authors describe pruning as a three-step pipeline—(1) **pre-training**, (2) pruning, and (3) retraining—but the term “pre-training” is potentially misleading here. It would be clearer to simply refer to “training, pruning, and retraining.”

2. **Discrepancy between theory and experimental setup:** The paper introduces the theoretical framework of oracle pruning (Eq. 1) and its Taylor approximation (Eq. 2).

    - However, the implementation described in Section 3.1 appears to approximate oracle pruning via a random-search strategy, making the connection between the theoretical formulation and the experimental procedure unclear.
    - Moreover, Section 3.3.5 and Table 3 report UMP and GMP, which are conventional magnitude-based pruning methods, but their connection to the oracle-pruning objective in Eqs. (1)–(2) is unclear.
    - In the experiments, it seems that calibration data is not required, as I did not see it discussed in the paper.

    I am very confused. Perhaps I am unfamiliar with common practice in CV settings; could the authors clarify more details?

3. **Clarification on the Research Objective**: I think the paper should not question the validity of oracle pruning, but rather the effectiveness of oracle pruning under the _retraining_ paradigm.

     - In my understanding, oracle pruning is primarily intended for **_pruning after training_**. Moreover, in Section 5, the authors acknowledge that oracle pruning can remain valid in certain contexts—particularly when retraining is not involved. Therefore, I think retraining is not a necessary component of pruning, but an additional step adopted in some works to improve performance.
     - If retraining is treated as mandatory, then the pruning criterion should account for the subsequent retraining stage; however, this setting is arguably closer to **_pruning during training_**, which is not discussed by Section 2.

      I find the overall research motivation and framing of the paper insufficiently justified.

**Audience:**

Yes

**Audience Explanation:**

I think some researchers will be interested in the experimental results presented in this paper.

**Broader Impact Concerns:**

I have no concerns about broader impact.

**Claims And Evidence:**

No

**Claims Explanation:**

I think this paper makes bold statements unsupported by rigorous evidence, and isn’t clearly written.

**Requested Changes:**

I have listed the requested changes in the weakness section. Please reference it.

---

> ### Author Response · Authors · 2025-12-23
> **Response to Reviewer cYmH (Part-1)**
>
> We sincerely thank the reviewer for the helpful comments regarding the clarity of our work. We address the questions one by one as follows, with the updated contents highlighted in **blue**.
>
> ---
>
> > **Q1:** *Suggestion on clarity and terminology.*
>
> **A:** We sincerely thank the reviewer for the helpful advice about the clarity of our work. We respond to the reviewer's concerns one by one as follows.
>
> - For Eq. (1), it formalizes the core idea of oracle pruning: among all pruning masks that retain exactly \(C\) parameters, the objective is to identify the one that minimizes the resulting increase in training loss. Here, \(M\) denotes a binary pruning mask, where each element indicates whether the corresponding weight in \(W\) is preserved (\(M_i = 1\)) or removed (\(M_i = 0\)), and the pruned parameters are obtained via element-wise masking \(W' = W \odot M\). We additionally add parentheses to clarify the scope of the minimization.
> - Following the reviewer's advice, we revise Eq. (2) to follow the standard exposition adopted in prior pruning literature (*e.g.*, OBS [`R1`] and LLM-Pruner [`R2`]) by expressing the second-order Taylor approximation in a unified quadratic form, without explicitly separating the diagonal and off-diagonal Hessian terms.
> - Regarding the terminology of the three-step pruning pipeline, we revise the manuscript to replace `pre-training` with `training` in line with the reviewer’s suggestion. We note that the term `pre-training` was originally used to distinguish the initial training stage prior to pruning from the subsequent retraining stage, following common usage in the pruning literature (*e.g.*, LLM-Pruner [`R2`] and DepGraph [`R3`]). To prevent potential confusion, we now adopt the revised terminology.
>
> We have updated our manuscript (Abstract, Section `1`, and Figure `1`) based on the above feedback.
>
> ---
>
> > **Q2:** *Suggestion on aligning theory with experimental practice.*
>
> **A:** We sincerely thank the reviewer for the constructive feedback. We respond to the reviewer's questions one by one as follows.
>
> - Oracle pruning aims to minimize the increase in training loss after pruning, implicitly assuming that a smaller training-loss increase correlates with better final performance after retraining. To examine the validity of this assumption, an exhaustive search over all pruning masks is needed, but infeasible. As a remedy, we adopt random pruning to sample a diverse set of pruning combinations, which allows us to empirically study the correlation between pruned train loss and final performance. Our analyses suggest that, in many cases, training loss after pruning is not reliably aligned with final performance (see Section `3.3`), thereby questioning the practical effectiveness of oracle pruning under retraining.
> - For TinyLLaVA-3.1B (Section `3.3.5`), we include several conventional baselines (*e.g.*, UMP and GMP) to compute corresponding pruned train loss and final performance. These methods are not intended to approximate the oracle-pruning objective in Eqs. (1)–(2); rather, they serve as representative baselines to provide additional data points for counterexample analysis (see details in Section `3.2`).
> - Our original experiments do **not** require additional calibration data, as they do not involve computing Hessian terms; all evaluations are conducted using the training and test sets of the datasets mentioned (*e.g.*, CIFAR and ImageNet). During the rebuttal, we further include supplementary experiments on Taylor-FO pruning in response to Reviewer `G7Rz`. In these experiments, a subset of CIFAR-100 (20k samples) is randomly selected as calibration data to compute the Taylor-FO scores.
>
> We have added more details to clarify our experiments (see details in Section `3` and Appendix) in the manuscript.

---

> ### Author Response · Authors · 2025-12-23
> **Response to Reviewer cYmH (Part-2)**
>
> > **Q3:** *Suggestion on refining research objective and framing*
>
> **A:** We sincerely thank the reviewer for the insightful suggestion. Our work revisits the effectiveness of oracle pruning under the retraining paradigm, which motivates the overall study. Following the reviewer’s comment, we further clarify and emphasize this motivation in Section `1` to avoid potential ambiguity or misleading interpretation.
>
> Regarding the comment *“the pruning criterion should account for the subsequent retraining stage”*, we fully agree with the reviewer. However, we note that the majority of existing pruning methods still design pruning criteria solely based on the pruning stage itself, without explicitly modeling retraining effects. To explore this gap, we include exploratory experiments in Appendix `D`. Our preliminary results indicate that model performance after full retraining is highly correlated with the performance obtained after only 10% retraining, suggesting that early retraining behavior may serve as a useful proxy. We believe this insight may inspire future work to incorporate retraining-aware signals into pruning criteria.
>
> Concerning the reviewer’s mention of *pruning during training*, we further respond as follows.
>
> 1. If *pruning during training* is interpreted as iterative pruning (*e.g.*, [`R4`]), we provide experiments in Appendix `B` on ResNet-50 with CIFAR-100 under an iterative pruning regime. We applied 5 pruning rounds, each removing 10% of the parameters (50% in total). The first four rounds were followed by 15 epochs of retraining, and the final round by 60 epochs, yielding 120 retraining epochs in total—identical to the one-shot pruning schedule used in the main paper. Iterative pruning provides intermediate pruned checkpoints, allowing us to record the train loss after each pruning step. We then computed the correlation between the pruned train losses and the final test performance under two different learning-rate schedules. Across both settings, we consistently observed no significant positive correlation between pruned train loss and final test loss, and no significant negative correlation between pruned train loss and final test accuracy. These findings further support our claim under the iterative pruning setting.
>
> *LR1: (first four iterations: lr = 0.001; final iteration: 60 epochs with lr schedule (0: 0.1, 20: 0.01, 40: 0.001))*
> | Iteration | Pruned Train Loss vs Final Test Loss (τ, p) | Pruned Train Loss vs Final Test Acc (τ, p) |
> |-|-|-|
> | 1/5 | (0.05, 0.31)  | (0.03, 0.58) |
> | 2/5 | (-0.05, 0.32) | (0.04, 0.47) |
> | 3/5 | (0.02, 0.71)  | (0.04, 0.41) |
> | 4/5 | (0.10, 0.06)  | (0.08, 0.13) |
> | 5/5 | (0.05, 0.34)  | (0.03, 0.52) |
>
> *LR2: (first four iterations: 15 epochs with lr schedule (0:0.1, 5:0.001, 10:0.0001); final iteration: 60 epochs with lr schedule (0: 0.1, 20: 0.01, 40: 0.001))*
> | Iteration | Pruned Train Loss vs Final Test Loss (τ, p) | Pruned Train Loss vs Final Test Acc (τ, p) |
> |-|-|-|
> | 1/5 | (-0.04, 0.48) | (-0.03, 0.64) |
> | 2/5 | (-0.06, 0.28) | (0.003, 0.95) |
> | 3/5 | (-0.12, 0.02) | (0.03, 0.55)  |
> | 4/5 | (-0.08, 0.14) | (0.07, 0.19)  |
> | 5/5 | (0.02, 0.72)  | (0.07, 0.22)  |
>
> 2. We also acknowledge a separate line of work that incorporates sparsification directly into the training process (refer as *pruning during training*), producing sparse networks after training. For example, [`R5`] proposes generalized dropout with learnable dropout rates, and [`R6`] addresses optimization challenges in training networks with L0 regularization. While related in spirit, such approaches typically fall outside the conventional scope of pruning, whose primary target remains pre-trained models. Our work likewise focuses on pruning pre-trained networks. Following the reviewer’s suggestion, we add a brief discussion of *pruning during training* in Section `2` to clarify these distinctions and contextualize our setting.
>
> We have updated our manuscript (Section `1`, Section `2`, and Appendix) according to the above responses.

---

> ### Author Response · Authors · 2025-12-23
> **Response to Reviewer cYmH (References)**
>
> **References:**
>
> - [`R1`] Hassibi, B., & Stork, D. (1992). Second order derivatives for network pruning: Optimal brain surgeon. Advances in neural information processing systems, 5.
> - [`R2`] Ma, X., Fang, G., & Wang, X. (2023). Llm-pruner: On the structural pruning of large language models. Advances in neural information processing systems, 36, 21702-21720.
> - [`R3`] Fang, G., Ma, X., Song, M., Mi, M. B., & Wang, X. (2023). Depgraph: Towards any structural pruning. In Proceedings of the IEEE/CVF conference on computer vision and pattern recognition (pp. 16091-16101).
> - [`R4`] Roy, S., Panda, P., Srinivasan, G., & Raghunathan, A. (2020, July). Pruning filters while training for efficiently optimizing deep learning networks. In 2020 International Joint Conference on Neural Networks (IJCNN) (pp. 1-7). IEEE.
> - [`R5`] Louizos, C., Welling, M., & Kingma, D. P. (2017). Learning sparse neural networks through L_0 regularization. arXiv preprint arXiv:1712.01312.
> - [`R6`] Srinivas, S., & Babu, R. V. (2016). Generalized dropout. arXiv preprint arXiv:1611.06791.
>
> ---
>
> Last but not least, we would like to sincerely thank Reviewer `cYmH` again for the valuable time and constructive feedback provided during this review.
>
> **Thank you for helping improve our work so far! We are actively available during the discussion period. Let us know should you have any further questions.**

---

### Review · Reviewer_GwpC · 2026-01-03

**Summary Of Contributions:**

The paper investigates the validity of oracle pruning -- that selecting weights to minimize the pruned training loss gives the best performance after retraining. Via a large-scale study of models, the authors suggest that oracle pruning holds for toy scenarios (small networks on MNIST) but fails for modern deep networks on CIFAR-10 and ImageNet. They argue that task complexity is the reason for this failure, and suggest that pruning criteria should consider retraining dynamics.

**Audience:**

Yes

**Audience Explanation:**

Despite the suspected flaw in the experimental methodology, I think it's still a clear and well-defined piece of research that some of TMLR's audience would find interesting. Challenging oracle pruning is an interesting question.

**Broader Impact Concerns:**

None.

**Claims And Evidence:**

No

**Claims Explanation:**

The paper is quite clear and cleanly presents a lot of empirical evidence for their claim. I would have found a linear regression analysis more compelling than reporting correlation coefficients (including visualized trend lines). I think the empirical setup is basically fine, and better than the average deep learning paper.

However, there may be a critical confounding factor that isn't super clear in the paper. Between the "toy" and "real" baselines, the authors also change from _exact_ oracle pruning to _approximate_ oracle pruning, simply due to the growing size of the search space. On the MNIST/LeNet experiments, I think the authors actually try all masks and choose the best one (oracle pruning). On larger experiments, it seems they sample ~1K masks and choose the best one. This is a radically different setup -- given the structure of the weight space it's unlikely you'd choose a "good" mask with just ~1K samples.

There are related ablations in Fig 5 and Fig A6. They show that increasing the width/depth of LeNet destroys correlation. But is this just because the search space for the mask gets larger and the authors switch from exact to approximate oracle pruning? I understand that doing the exact search may be intractable, so a good baseline to include might be approximate search on the toy examples.

I'm assuming that when people talk about oracle pruning, they're thinking of the exact thing, not the approximate thing -- and papers are based on it insofar as they try to find better-than-random approximations of the real mask. I'm not a researcher in the area of pruning, so I could be wrong here.

What I would find more convincing to investigate oracle pruning is something like: increasing search time (ideally not _random_ search) for the mask does not increase test accuracy after retraining (presented, say, against log-scale search time).

**Requested Changes:**

- Can you apply the random sampling to the small LeNet baseline?
- Consider moving Figure A6 to the main text, I thought this was a useful ablation and helpful alongside Figure 5
- Do you have an argument for why random search of ~1K masks is a good approximation to oracle pruning?

---

> ### Author Response · Authors · 2026-01-11
> **Response to Reviewer GwpC (Part-1)**
>
> We sincerely thank the reviewer for the constructive comments and for acknowledging the clarity of our work. We address each concern in turn below, with revisions highlighted in **blue**.
>
> ---
>
> We notice that a central concern raised by the reviewer under `Are the claims made in the submission supported by accurate, convincing, and clear evidence?` is the implementation of oracle pruning across different architectures.
>
> > **Q:** *Suggestion on clarity for oracle pruning implementation.*
>
> **A:** We thank the reviewer for the constructive feedback and fully acknowledge the concern regarding the implementation of oracle pruning. Our responses are as follows.
>
> 1. The core idea of oracle pruning is to assume that, after pruning a group of parameters, a smaller increase in training loss indicates a higher expected final performance. Ideally, oracle pruning corresponds to identifying the pruning configuration that minimizes the increase in training loss.
>
> 2. Following the above point 1, the primary goal of our experiments is to examine the validity of the core principle underpinning oracle pruning, rather than to demonstrate that our random sampling can cover the oracle pruning solution.
>
> We fully agree with the reviewer on that, when the search space becomes larger, exhaustively searching the true oracle pruning result is infeasible. This observation is precisely aligned with our experimental motivation: given the prohibitively large search space, instead of attempting to searching the true oracle pruning solution, we adopt a tractable approach to evaluate the oracle pruning principle. Specifically, by randomly sampling a sufficient number of pruning combinations, we can reliably estimate the correlation between the two key quantities underlying the oracle pruning idea (**pruned train loss** & **final performance**). The correlation directly indicates the validity of the oracle pruning principle.
>
> 3. Regarding the reviewer's question on exact and approximate oracle pruning, our choice of implementation is primarily driven by our focus on correlation analysis, rather than by approximating oracle pruning within a given search space. When the search space is small (*e.g.*, 45 pruning combinations for Conv1 pruning with a pruning ratio of 0.8 on LeNet5), a limited number of data points are available to compute correlations. For larger search spaces, sampling a sufficient number of pruning combinations (*e.g.*, ~1K) provides enough data points to yield a *statistically reliable* correlation estimation.
>
> 4. Considering the suggestion of more convincing oracle pruning by the reviewer, in the pruning community, some pruning methods (*e.g.*, Taylor-FO pruning) are often used to approximate oracle pruning instead of random pruning with all combinations. Under these approximation methods [`R1`, `R2`], the focus is often on how to achieve a better approximation, rather than scaling the search time. Furthermore, we conduct experiments using Taylor-FO pruning on ResNet50 (CIFAR100), and the results also show that there is no significant correlation between pruned train loss and final performance (see Appendix `B.1`).
>
> *Pruning ratio = 0.5, ResNet50 with CIFAR100*
> | Method | Pruned Train Loss vs Final Test Loss (τ, p) | Pruned Train Loss vs Final Test Acc (τ, p) |
> |-|-|-|
> | Taylor-FO pruning | (-0.009, 0.86) | (0.046, 0.39) |
>
> 5. Furthermore, we revisited the manuscript and identified places where the writing could be misleading. We have revised the text accordingly. For example, in the caption of Figure `2`, we clarified that the blue star denotes a reference oracle pruning point, defined as the one with the smallest pruned train loss among all evaluated pruning combinations.
>
> We have updated our manuscript (Figure `2`, Section `3`, and Appendix `B.1`) according to the above responses.
>
> ---
>
> > **RC1:** *Question about random pruning on LeNet5.*
>
> **A:** We sincerely thank the reviewer for the useful suggestion. In our current experiments on LeNet-5, we select multiple pruning combinations (*i.e.*, when the search space is small, we use all pruning combinations; when it is large, we sample a sufficiently large number (~1K) of combinations). In this context, random pruning refers to sampling a pruning combination from the set of all possible combinations. Therefore, our experiments are actually based on random pruning. The key difference is that we perform random pruning at scale and leverage the resulting data to compute correlations, which enables us to systematically assess the validity of oracle pruning.
>
> ---
>
> > **RC2:** *Suggestion for moving figure to the main text.*
>
> **A:** We sincerely thank the reviewer for the helpful suggestion. Following the reviewer's suggestion, we have moved Figure `A6` and updated the corresponding context in the updated manuscript (Section `4`).

---

> > ### Author Response · Authors · 2026-01-11
> > **Response to Reviewer GwpC (Part-2)**
> >
> > > **RC3:** *Question about approximation to oracle pruning.*
> >
> > **A:** We sincerely thank the reviewer for the insightful suggestion. We would like to clarify that performing random pruning for ~1K trials is not intended to approximate oracle pruning. As we mentioned in the above rebuttal to `Q`, the primary goal of our experiments is to examine the core idea of oracle pruning (*i.e.*, after pruning a group of parameters, a smaller increase in training loss indicates a higher expected final performance) instead of using random search to approximate oracle pruning.
> >
> > We have revisited our writing and updated some context, which may be confusing in our manuscript (Section `3`).
> >
> > ---
> >
> > **References:**
> >
> > - [`R1`] Molchanov, Pavlo, et al. "Pruning convolutional neural networks for resource efficient inference." arXiv preprint arXiv:1611.06440 (2016).
> > - [`R2`] Ma, X., Fang, G., & Wang, X. (2023). Llm-pruner: On the structural pruning of large language models. Advances in neural information processing systems, 36, 21702-21720.
> >
> > ---
> >
> > Last but not least, we would like to sincerely thank Reviewer `GwpC` again for the valuable time and constructive feedback provided during this review.
> >
> > **Thank you for helping improve our work so far! We are actively available during the discussion period. Let us know should you have any further questions.**

---

### Decision · Action_Editor_xdBZ · 2026-06-05

**Recommendation:** Accept with minor revision

**Additional Comments:**

Please modify the abstract to make the title-paper gap more obvious.  I do not see how the title can be modified; but if the title can be modified to refelct the gap, that will be even better.

**Audience:**

Yes

**Audience Explanation:**

This paper provides a relatively thorough examination of oracle pruning—a long-standing assumption in model compression. The model compression community will find this work deeply interesting and methodologically instructive.

**Claims And Evidence:**

Yes

**Claims Explanation:**

The authors claim that using oracle pruning, the performance before and after retraining has a small correlation. The claim is supported by extensive experiments with a wide range of models, datasets, and pruning ratios, with 37K runs. The reviewers pointed out a discrepancy that the authors used approximate oracle pruning due to the computational challenge of exact oracle pruning. The authors modified the wording a bit, and the reviewers agree that the claim is validated, after the modification. Nevertheless, reviwers still think the title sets up the expectation of seeing a different paper with exact oracle. I suggest the authors change the abstract to mention that for larger experiments, approximate oracle is used.